# When Shared Knowledge Hurts: Spectral Over-Accumulation in Model Merging

**Yayuan Li** [1] **Ze Peng** [1] **Jian Zhang** [1] **Jintao Guo** [1,2] **Yue Duan** [1] **Yinghuan Shi** [1,2]

## Abstract

Model merging combines multiple fine-tuned models into a single model by *adding* their weight updates, providing a lightweight alternative to re-training. Existing methods primarily target resolving conflicts between task updates, leaving the failure mode of over-counting shared knowledge unaddressed. We show that when tasks share aligned spectral directions (*i.e.*, overlapping singular vectors), a simple linear combination repeatedly accumulates these directions, inflating the singular values and biasing the merged model toward shared subspaces. To mitigate this issue, we propose Singular Value Calibration (SVC), a training-free and data-free post-processing method that quantifies subspace overlap and rescales inflated singular values to restore a balanced spectrum. Across vision and language benchmarks, SVC consistently improves strong merging baselines and achieves state-of-the-art performance. Furthermore, by modifying only the singular values, SVC improves the performance of Task Arithmetic by 13.0%. Code is available at https://github.com/lyymuwu/SVC.

## 1. Introduction

Model merging combines trained models into a single model by operating directly in weight space (Ruan et al., 2025). Compared with retraining from scratch or classical ensembling, it enables direct manipulation of weight updates to integrate capabilities (Shah et al., 2025; Ortiz-Jimenez et al., 2023), forget undesirable knowledge (Ilharco et al., 2022; Ni et al., 2024), and accelerate iteration of large-scale models (including LLMs) (Goddard et al., 2024; Wan et al., 2024). Current research focuses on merging models fine-tuned on

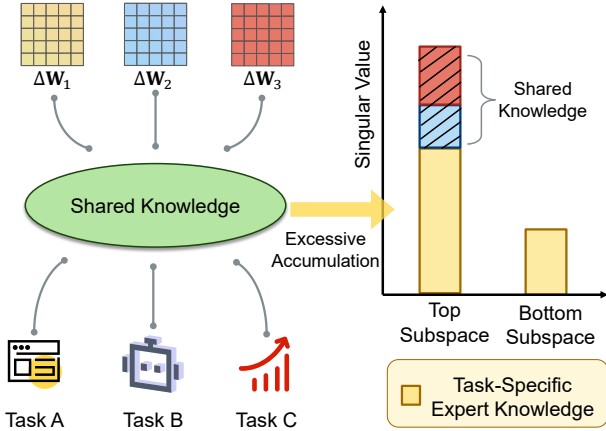

*Figure 1.* **Shared knowledge accumulation in model merging.** When merging task matrices ($\Delta\mathbf{W}_i$) from multiple tasks, shared knowledge that aligns across tasks can be over-counted, resulting in singular-value inflation in the merged model's spectrum. This inflation is concentrated in a few top spectral subspaces, causing the merged model to be dominated by shared directions, while task-specific components in the remaining subspaces are suppressed.

different tasks using the same pre-trained backbone, yielding a single model with enhanced multi-task capabilities. A central tool in this process is the use of task vectors (Ilharco et al., 2022), which capture differences between pre-trained and fine-tuned weights. In this context, the matrix-valued weight difference at each layer is referred to as the task matrix (TM) (Marczak et al., 2025a). Task matrices play a crucial role in analyzing interference during merging.

Prior work mainly improves merging by mitigating task-matrix conflicts. However, conflict is not the only source of interference. Recent studies report that cross-task alignment in spectral directions, manifested as overlap among singular vectors of different task matrices, is also associated with degradation after merging (Gargiulo et al., 2025). At first glance, this is unexpected. Components shared across tasks, which we call shared knowledge, are typically viewed as useful signals that should transfer rather than harm.

This observation points to a complementary mechanism that governs merged performance. A task is influenced not only by its task-specific update, but also by shared knowledge introduced by other tasks through the merge (Marczak et al.,

[1]National Key Laboratory for Novel Software Technology, Nanjing University, Nanjing 210023, China. [2]Institute of Brain-Computer Interface, Nanjing University, Nanjing 210023, China.. Correspondence to: Yinghuan Shi <syh@nju.edu.cn>.

*Proceedings of the 43ʳᵈ International Conference on Machine Learning*, Seoul, South Korea. PMLR 306, 2026. Copyright 2026 by the author(s).

2025a; Sun et al., 2025b). When several task matrices align in the same spectral subspaces, a simple linear combination repeatedly aggregates the same shared spectral components, leading to spectral over-counting in those subspaces. As a consequence, the merged update exhibits singular-value inflation and an imbalanced spectrum. Crucially, this inflation concentrates in a few top spectral subspaces with the largest singular values, so the merged model overemphasizes these dominant directions while underrepresenting the remaining subspaces, reducing downstream performance. Fig. 1 visualizes this pattern: shared knowledge is widespread across tasks, yet its over-counting leads to inflation in a few top spectral subspaces. These findings motivate measuring subspace-wise over-counting and correcting inflated singular values during merging.

To address this issue, we propose Singular Value Calibration (SVC), a training-free and data-free method for calibrating a merged update from any existing merging method in spectral space. SVC targets the singular-value inflation caused by spectral over-counting, so that the merged model is less dominated by a few shared subspaces.

To make this correction possible, we decompose the merged task matrix into spectral subspaces and use its output-space basis as a shared coordinate system. On this basis, each task matrix can be expressed as subspace-wise responses, which makes different tasks comparable within the same subspace. With tasks aligned to the same basis, we evaluate whether the merged update preserves each task-specific response. Specifically, we project the merged response onto the task-specific response and use the resulting projection coefficient to quantify how much the merged update amplifies that direction. A coefficient larger than $1$ means the merged update has accumulated extra mass along that direction, indicating spectral over-counting from cross-task alignment.

Based on these projection coefficients, SVC measures subspace-wise overlap and converts it into a calibration strength for each spectral subspace. It then rescales the corresponding singular values while keeping the spectral directions unchanged. As a result, SVC restores a more balanced spectrum and consistently improves merging performance across vision and language benchmarks. We summarize our contributions as follows:

- We identify spectral over-counting as a key failure mode in model merging, where redundant aggregation of shared knowledge induces singular-value inflation in a small set of dominant spectral subspaces and suppresses other components.

- We propose Singular Value Calibration (SVC), a training-free and data-free method that quantifies output-space overlap in the merged spectral basis and calibrates singular values to restore spectral balance.

- We provide theoretical analysis and empirical evidence validating SVC, which improves Task Arithmetic by 13.0%, and enables targeted improvements for specific tasks such as preference optimization.

## 2. Related Work

**Dynamic Model Merging.** Unlike model ensembling, which combines the outputs or predictions of multiple independent models to improve generalization (Dong et al., 2020), model merging operates directly at the weight level. In its common training-free (Zhang et al., 2025; Li et al., 2025a; Yuan et al., 2025d) form, it integrates the knowledge encoded in the parameters of several trained models into a single unified model (Yang et al., 2024a). This approach addresses challenges such as catastrophic forgetting (Chitale et al., 2023; Zhu et al., 2024; Marczak et al., 2025b), domain shift (Izmailov et al., 2019; Wortsman et al., 2022), and the efficient construction of LLMs (Dekoninck et al., 2024; Aiello et al., 2023). To reduce conflicts among models, a line of work studies dynamic merging, where the behavior of the merged model depends on the input. For example, DaWin (Oh et al., 2024) performs input-conditioned interpolation, EMR-Merging (Huang et al., 2024) and TALL-Mask (Wang et al., 2024) learn task-specific masks, and Twin-Merging (Lu et al., 2024) introduces task experts in the spirit of mixture-of-experts. These approaches can be effective, but they require task labels or routing decisions at inference time; in contrast, SVC is a static, training- and data-free post-hoc calibration that improves merged models without any additional inference-time information.

**Static Model Merging.** Early research in model merging primarily focused on weight averaging and traditional interpolation strategies (Wortsman et al., 2022; Ilharco et al., 2022; Matena & Raffel, 2022; Jin et al., 2023). These methods allowed rapid assembly of models with expertise from multiple tasks but often resulted in sub-optimal performance due to unresolved conflicts or redundancies among weights. Subsequent work has attempted to address these issues by mitigating inter-model conflicts under either data-dependent or data-free settings. Data-dependent methods typically require auxiliary data, such as validation sets or unlabeled test inputs. For example, PCB-Merging (Du et al., 2024) uses a validation set, NPS-Pruning (Du et al., 2025) relies on a calibration set, and AdaMerging (Yang et al., 2024c) and Trust Region Arithmetic (Sun et al., 2025b) perform test-time adaptation. However, these methods are less practical in scenarios where auxiliary data is unavailable. Data-free approaches can be further categorized into weight-space and spectral-space methods. Weight-space methods (Yu et al., 2024; Yadav et al., 2023; He et al., 2024) focus on localizing and pruning conflicting parameters to reduce the interference between the tasks. However, parameter-level

corrections may not fully capture the structured accumulation of task knowledge across shared subspaces, motivating SVC to study model merging from a spectral-space perspective. However, such weight-space operations mainly treat parameters as independent local units, making it difficult to capture the global structural properties of task updates; therefore, further work, including our SVC, investigates model merging from a spectral-space perspective.

**Spectral-Space Merging.** Recent studies have explored model merging from a spectral-space perspective by applying singular value decomposition (SVD) to task updates. TSV (Gargiulo et al., 2025) decomposes task matrices into singular directions to orthogonalize task-specific components and reduce interference. STAR (Lee et al., 2025) performs spectral truncation and rescaling to remove less informative components while preserving task-update scale. Iso-C and Iso-CTS (Marczak et al., 2025a) study the alignment between task-specific and merged singular subspaces, and improve merging by globally adjusting singular values in the spectrum. Subspace-Boosting (Skorobogat et al., 2025) further identifies rank collapse in merged task-vector spaces and boosts underutilized spectral dimensions. Different from these methods, which mainly use SVD to construct a better merged update through selecting, pruning, or reweighting spectral components, SVC is a post-hoc calibration method. Given an already merged update $\Delta W_{\text{merge}}$, diagnoses task contributions in each merged output subspace and adaptively corrects the subspace-wise singular-value imbalance without redesigning the merging rule or modifying the merged singular vectors.

## 3. Spectral View of Inter-Model Interference

This section answers a single question: why does merging multiple task updates hurt performance even when tasks appear aligned? Our goal is to isolate an interference source that is not explained by weight conflicts. To do so, we analyze merged updates in spectral space and track how shared components accumulate across tasks. We show that repeated accumulation can over-count shared directions and inflate singular values in a few dominant subspaces.

### 3.1. Preliminary

Given a set of $K$ fine-tuned model parameter sets $\{\mathbf{W}_i\}_{i=1}^{K}$, each obtained by fine-tuning the pre-trained parameter $\mathbf{W}_{\text{pre}}$. Model merging aims to construct a merged model $\mathbf{W}_{\text{merge}}$ that effectively inherits task-specific knowledge from all given $\{\mathbf{W}_i\}_{i=1}^{K}$. Traditionally, a simple weight averaging is adopted: $\mathbf{W}_{\text{merge}} = \frac{1}{K} \sum_{i=1}^{K} \mathbf{W}_i$. Building on this, Task Arithmetic (TA) (Ilharco et al., 2022) rewrites merging in terms of task updates. For a given layer, the task matrix for task $i$ is $\Delta \mathbf{W}_i = \mathbf{W}_i - \mathbf{W}_{\text{pre}}$. A merging method

then combines $\{\Delta \mathbf{W}_i\}_{i=1}^{K}$ to produce a merged task matrix $\Delta \mathbf{W}_{\text{merge}}$. The final merged model is obtained by adding the merged update back to the pre-trained weights, with a global scaling $\lambda$:

$$\mathbf{W}_{\text{merge}} = \mathbf{W}_{\text{pre}} + \lambda \Delta \mathbf{W}_{\text{merge}}. \quad (1)$$

### 3.2. Projections Reveal Spectral Over-Counting

Many existing merging methods aim to make the merged model approximate the behavior of expert models (*i.e.*, task-specific model) (Li et al., 2025b). However, there lack of an effective way to measure such a gap in a data-free setting.

To address this issue, we introduce an output-space projection-based interference metric in this subsection.

**Projecting Tasks onto Output-Space.** Consider a merged task matrix $\Delta \mathbf{W}_{\text{merge}} = \sum_{i=1}^{K} \Delta \mathbf{W}_i$ and its Singular Value Decomposition (SVD):

$$\Delta \mathbf{W}_{\text{merge}} = \mathbf{U}\mathbf{\Sigma}\mathbf{V}^{\top}, \qquad \mathbf{\Sigma} = \text{diag}(\sigma^1, \ldots, \sigma^R), \quad (2)$$

where $\boldsymbol{u}^r$ and $\boldsymbol{v}^r$ are the $r$-th left and right singular vectors. We use the index $r$ to label a spectral subspace in matrix space, *i.e.*, the subspace spanned by the $\boldsymbol{u}^r(\boldsymbol{v}^r)^{\top}$.

For a task matrix $\Delta \mathbf{W}_i \in \mathbb{R}^{m \times n}$, its response to an input $\boldsymbol{x} \in \mathbb{R}^n$ is

$$\boldsymbol{y}_i(\boldsymbol{x}) = \Delta \mathbf{W}_i \boldsymbol{x} \in \mathbb{R}^m. \quad (3)$$

To analyze the behavior of $\Delta \mathbf{W}_i$ along the output-space direction $\boldsymbol{u}^r$, we project this response onto $\boldsymbol{u}^r$:

$$(\boldsymbol{u}^r)^{\top} \boldsymbol{y}_i(\boldsymbol{x}) = (\boldsymbol{u}^r)^{\top} \Delta \mathbf{W}_i \boldsymbol{x}. \quad (4)$$

Therefore, we define

$$\boldsymbol{a}_i^r := (\boldsymbol{u}^r)^{\top} \Delta \mathbf{W}_i \in \mathbb{R}^n, \quad (5)$$

which acts as a data-independent descriptor of how $\Delta \mathbf{W}_i$ responds along $\boldsymbol{u}^r$. In other words, for any input $\boldsymbol{x}$, $\boldsymbol{a}_i^r \boldsymbol{x}$ gives the scalar response of $\Delta \mathbf{W}_i \boldsymbol{x}$ projected onto $\boldsymbol{u}^r$. Thus, $\boldsymbol{a}_i^r$ characterizes the behavior of task $i$ along the output-space direction $\boldsymbol{u}^r$ in a data-free manner.

*Remark* 3.1 (Layer-wise linear view). We treat the task matrix as a local linear operator within a layer (or block) and study how task matrices mix along output-space directions.

**Interference as Output-Space Behaviour Inconsistency.** Having projected task matrix onto the shared output-space basis, we can now quantify inter-task interference within each spectral subspace. Specifically, we form the merged response in subspace $r$ by summing the corresponding subspace responses $\boldsymbol{a}_i^r$:

$$\boldsymbol{a}_{\text{merge}}^r := (\boldsymbol{u}^r)^{\top} \Delta \mathbf{W}_{\text{merge}} = \sum_{i=1}^{K} \boldsymbol{a}_i^r. \quad (6)$$

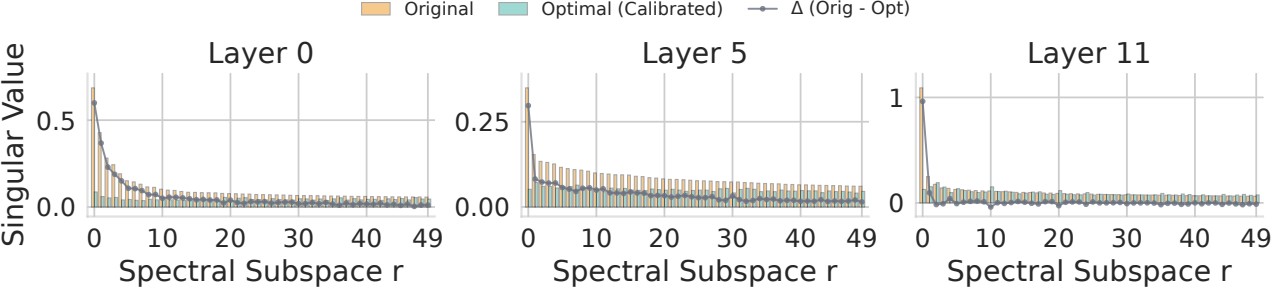

*Figure 2.* **Discrepancy between original and calibrated singular values.** We conduct this analysis on the 8-task CV classification merging setup. For weight-space addition, we compare the original singular values $\sigma$ from $\mathrm{SVD}(\Delta\mathbf{W}_{\mathrm{merge}})$ with the calibrated values $\sigma^\star$, where $\sigma^\star$ is obtained by first computing the task-wise optimal scalings $(\gamma_i^r)^\star$ from Eq. (13) and then averaging them across tasks within each subspace. A clear gap $\Delta = \sigma - \sigma^\star$ appears in top spectral subspaces, indicating systematic spectral over-counting and singular-value inflation.

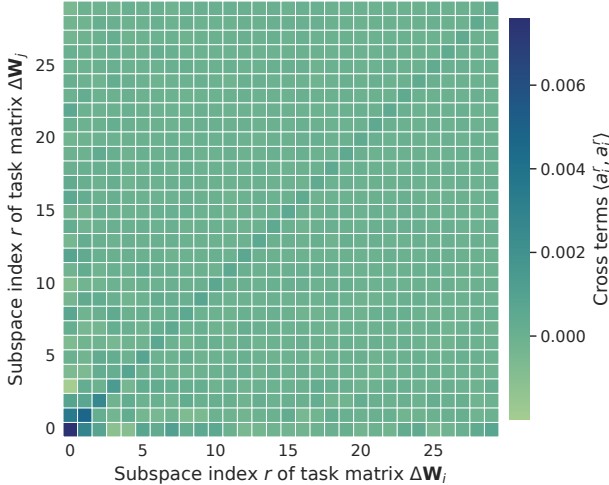

*Figure 3.* **Cross terms concentrate in top spectral subspaces.** We visualize the unnormalized cross-task inner products $\langle a_i^r, a_j^r \rangle$ for a small top subspace, showing predominantly positive overlap that will induces over-counting.

Ideally, if the merged task matrix fully preserved task $i$'s capability in subspace $r$, then the component of the merged response $\boldsymbol{a}_{\mathrm{merge}}^r$ along $\boldsymbol{a}_i^r$ should match $\boldsymbol{a}_i^r$ in magnitude. We therefore measure the interference $\mathcal{I}_i^r$ by the mismatch between task $i$'s subspace response $\boldsymbol{a}_i^r$ and the projection of the merged response $\boldsymbol{a}_{\mathrm{merge}}^r$ onto $\boldsymbol{a}_i^r$:

$$\mathcal{I}_i^r := \left\| \mathrm{Proj}_{\boldsymbol{a}_i^r}(\boldsymbol{a}_{\mathrm{merge}}^r) - \boldsymbol{a}_i^r \right\|_2^2 \geq 0, \qquad (7)$$

where

$$\mathrm{Proj}_{\boldsymbol{a}_i^r}(\boldsymbol{a}_{\mathrm{merge}}^r) \triangleq s_i^r \boldsymbol{a}_i^r = \frac{\langle \boldsymbol{a}_{\mathrm{merge}}^r, \boldsymbol{a}_i^r \rangle}{\|\boldsymbol{a}_i^r\|_2^2} \boldsymbol{a}_i^r. \quad (8)$$

The projection coefficient $s_i^r$ quantifies how the merged response scales task $i$ along $\boldsymbol{a}_i^r$ in subspace $r$: $s_i^r > 1$ indicates amplification, while $s_i^r < 1$ indicates attenuation

(or conflict). Consequently, $\mathcal{I}_i^r = 0$ holds if and only if $\mathrm{Proj}_{\boldsymbol{a}_i^r}(\boldsymbol{a}_{\mathrm{merge}}^r) = \boldsymbol{a}_i^r$, equivalently $s_i^r = 1$. Building on this projection coefficient, we derive the following lemma:

**Lemma 3.2** (Cross-term form of projection interference). *Assume* $\Delta\mathbf{W}_{\mathrm{merge}} = \sum_{k=1}^K \Delta\mathbf{W}_k$. *Fix any task $i$ and subspace $r$, and assume* $\|\boldsymbol{a}_i^r\|_2^2 > 0$. *Then*

$$s_i^r = 1 + \sum_{j \neq i} \frac{\langle \boldsymbol{a}_j^r, \boldsymbol{a}_i^r \rangle}{\|\boldsymbol{a}_i^r\|_2^2}, \qquad \mathcal{I}_i^r = (s_i^r - 1)^2 \|\boldsymbol{a}_i^r\|_2^2. \quad (9)$$

Lemma 3.2 makes the source of projection mismatch explicit. **Projection mismatch is governed by the cross-task inner products** $\langle \boldsymbol{a}_j^r, \boldsymbol{a}_i^r \rangle$, **which quantifies how strongly other tasks contribute along task $i$'s response direction in subspace $r$.** When many $\langle \boldsymbol{a}_j^r, \boldsymbol{a}_i^r \rangle$ terms are positive, we obtain $s_i^r > 1$ and thus $\mathcal{I}_i^r > 0$, meaning that multiple tasks jointly accumulate along the same direction $\boldsymbol{a}_i^r$ and the merged response over-counts this shared component. This concentration of cross-task overlap is visible in Fig. 3, where large overlaps cluster in the top spectral subspaces.

**From Interference to Singular-Value Inflation.** The projection mismatch above is a behavioral symptom; to correct it, we next trace its impact back to the parameters.

By Eq. (2), the merged response in subspace $r$ satisfies $\boldsymbol{a}_{\mathrm{merge}}^r = \sigma^r (\boldsymbol{v}^r)^\top$, hence the singular value is exactly the response magnitude:

$$\sigma^r = \left\| \boldsymbol{a}_{\mathrm{merge}}^r \right\|_2. \quad (10)$$

Thus, once we characterize how projection mismatch changes the magnitude of $\boldsymbol{a}_{\mathrm{merge}}^r$, we can directly translate it into a statement about the singular value $\sigma^r$. This connection yields the following theorem.

**Theorem 3.3** (Projection-optimal calibration and singular–value inflation). *Fix a target task $i$ and a subspace $r$, and*

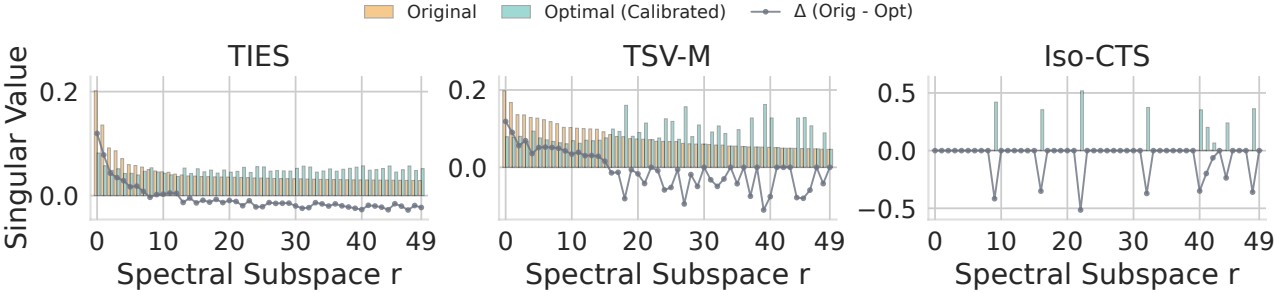

*Figure 4.* **Generality of the singular-value gap.** We conduct this analysis on the 8-task CV classification merging setup and compare the original singular values $\sigma$ with the calibrated values $\sigma^\star$, where $\sigma^\star$ applies the subspace-wise average of the task-wise optimal scalings $\gamma_i^{r\,\star}$ from Eq. (13).The gap between $\sigma$ and $\sigma^\star$ persists across representative merging methods, indicating that singular-value inflation and overly small singular values coexist.

assume $\|\boldsymbol{a}_i^r\|_2^2 > 0$. *Let $\boldsymbol{a}_{\mathrm{merge}}^r$ denote the merged response in subspace $r$. Consider the calibration problem*

$$\min_{\gamma^r \geq 0} \left\| \mathbf{Proj}_{\boldsymbol{a}_i^r}(\gamma^r \boldsymbol{a}_{\mathrm{merge}}^r) - \boldsymbol{a}_i^r \right\|_2^2. \qquad (11)$$

*Define $s_i^r$ as in Eq. (8). If $s_i^r > 0$, then the optimal calibration has the closed form*

$$(\gamma_i^r)^\star = \frac{1}{s_i^r} = \frac{\|\boldsymbol{a}_i^r\|_2^2}{\langle \boldsymbol{a}_{\mathrm{merge}}^r, \boldsymbol{a}_i^r \rangle}, \qquad (12)$$

*whereas if $s_i^r \leq 0$, the optimum is attained at the boundary $(\gamma_i^r)^\star = 0$. If $s_i^r > 0$, the corresponding projection-optimal singular value for best emulating task $i$ in subspace $r$ is*

$$(\sigma_i^r)^\star = (\gamma_i^r)^\star \sigma^r. \qquad (13)$$

*Under weight addition $\Delta\mathbf{W}_{\mathrm{merge}} = \sum_{k=1}^K \Delta\mathbf{W}_k$, if*

$$\sum_{j \neq i} \langle \boldsymbol{a}_j^r, \boldsymbol{a}_i^r \rangle > 0, \qquad (14)$$

*then $s_i^r > 1$ and thus $(\gamma_i^r)^\star < 1$. Equivalently,*

$$\sigma^r > (\sigma_i^r)^\star, \qquad (15)$$

*showing that positive cross-task overlap inflates the merged singular value above the projection-optimal magnitude.*

The condition $\langle \boldsymbol{a}_j^r, \boldsymbol{a}_i^r \rangle > 0$ implies that other tasks contribute constructively along task $i$'s direction in subspace $r$, increasing $\langle \boldsymbol{a}_{\mathrm{merge}}^r, \boldsymbol{a}_i^r \rangle$ and forcing $(\gamma_i^r)^\star < 1$ to match $\boldsymbol{a}_i^r$ under projection. **In particular, whenever $\sum_{j \neq i} \langle \boldsymbol{a}_j^r, \boldsymbol{a}_i^r \rangle > 0$, the merged response exhibits constructive accumulation, and thus the singular value $\sigma^r$ is inflated above the projection-induced optimal singular values $(\sigma_i^r)^\star$.** Such accumulation of singular values ultimately leads to a decline in model performance. Empirically, such positive overlap is most pronounced in leading subspaces (Fig. 3), yielding amplified top singular values in the merged spectrum (Fig. 2).

### 3.3. Generality of Singular-Value Over-Accumulation

So far, we have used weight addition to make the mechanism transparent. We next ask whether the same failure mode appears in other merging methods.

We answer this by comparing the original singular values $\sigma$ with the calibrated values $\sigma^\star$. Fig. 4 shows that the gap $\Delta = \sigma - \sigma^\star$ persists across representative methods. In many cases, the largest positive gaps still concentrate in the leading spectral subspaces, indicating over-accumulation of shared directions. At the same time, some methods exhibit large variance across the spectrum and can even yield negative gaps in certain subspaces, which corresponds to under-accumulation. Taken together, these observations suggest that most practical merging methods can simultaneously inflate dominant subspaces while shrinking others, motivating a calibration step that acts at the subspace level.

## 4. Methodology

To correct spectral over-counting in a merged update without additional data or optimization, we introduce **Singular Value Calibration (SVC)**, which post-processes a merged task matrix by adjusting its singular values while keeping the directions unchanged.

Given a pre-trained model $\mathbf{W}_{\mathrm{pre}}$ and $K$ fine-tuned models $\{\mathbf{W}_i\}_{i=1}^K$, we form task matrices:

$$\Delta\mathbf{W}_i = \mathbf{W}_i - \mathbf{W}_{\mathrm{pre}}. \qquad (16)$$

Let $\Delta\mathbf{W}_{\mathrm{merge}}$ be the merged task matrix produced by a base merging method (for example, summation, averaging, or masking). SVC takes $\Delta\mathbf{W}_{\mathrm{merge}}$ as input, estimates how much each spectral subspace is over-counted, and then rescales the corresponding singular values accordingly.

**Step 1: Merged Output-Space Basis.** We first choose a shared coordinate system so that different tasks can be compared within the same subspaces. To do so, we compute

the SVD of the merged task matrix

$$\Delta \mathbf{W}_{\text{merge}} = \mathbf{U}\boldsymbol{\Sigma}\mathbf{V}^\top = \sum_{r=1}^{R} \sigma^r \, \boldsymbol{u}^r (\boldsymbol{v}^r)^\top. \qquad (17)$$

We use the left singular vectors $\{\boldsymbol{u}^r\}$ as the merged output-space basis. This aligns with Section 3.2, where subspace-wise interactions are defined by projecting task matrices onto shared output-space directions.

**Step 2: Subspace-Wise Overlap from Projections.** With this basis fixed, we next quantify how much each subspace is over-counted after merging. For each subspace $r$ and each task $i$, we compute the task response and the merged response along $\boldsymbol{u}^r$:

$$\boldsymbol{a}_i^r = (\boldsymbol{u}^r)^\top \Delta \mathbf{W}_i \in \mathbb{R}^n, \ \boldsymbol{a}_{\text{merge}}^r = (\boldsymbol{u}^r)^\top \Delta \mathbf{W}_{\text{merge}}. \qquad (18)$$

We then measure how the merged response scales task $i$ along its own direction using the projection coefficient from Theorem 3.3:

$$s_i^r = \frac{\langle \boldsymbol{a}_{\text{merge}}^r, \boldsymbol{a}_i^r \rangle}{\|\boldsymbol{a}_i^r\|_2^2}. \qquad (19)$$

If multiple tasks contribute constructively in the same subspace, then $s_i^r > 1$, indicating over-counting.

To produce a single correction per subspace, we aggregate these coefficients across tasks into a calibration factor

$$\gamma^r = K / \sum_{i=1}^{K} \max(\alpha, s_i^r). \qquad (20)$$

Equivalently, $\gamma^r$ can be viewed as the harmonic mean of the clipped task-wise scalings $\{1/\max(\alpha, s_i^r)\}_{i=1}^K$. This choice is conservative: it down-weights subspaces primarily when many tasks exhibit large $s_i^r$ (strong over-counting), while $\alpha \in (0, 1]$ prevents unstable behavior when some $s_i^r$ are very small. In practice, $\gamma^r \approx 1$ indicates little systematic over-counting in subspace $r$, while $\gamma^r < 1$ indicates singular-value inflation.

**Step 3: Singular-Value Calibration and Reconstruction.** By Eq. (10), scaling $\boldsymbol{a}_{\text{merge}}^r$ is equivalent to scaling $\sigma^r$. Thus, singular-value inflation is corrected by rescaling each singular value based on the subspace-wise overlap degree.

$$\tilde{\sigma}^r = \gamma^r \sigma^r, \qquad (21)$$

With $\alpha$ applied inside $\gamma^r$, calibration is suppression-only when $\alpha = 1$, since $\max(\alpha, s_i^r) \geq 1$ makes $\gamma^r \leq 1$ for all $r$. Finally, we reconstruct the calibrated merged task matrix

$$\Delta \tilde{\mathbf{W}}_{\text{merge}} = \sum_{r=1}^{R} \tilde{\sigma}^r \, \boldsymbol{u}^r (\boldsymbol{v}^r)^\top, \qquad (22)$$

---

**Algorithm 1** Subspace-Consistency Spectral Calibration

**Input:** $\mathbf{W}_{\text{pre}}, \{\mathbf{W}_i\}_{i=1}^K, \Delta \mathbf{W}_{\text{merge}}, \alpha$.
**Output:** $\mathbf{W}_{\text{merge}}$
$\Delta \mathbf{W}_i \leftarrow \mathbf{W}_i - \mathbf{W}_{\text{pre}}$ for $i = 1, \dots, K$
$\Delta \mathbf{W}_{\text{merge}} = \mathbf{U}\boldsymbol{\Sigma}\mathbf{V}^\top$ (SVD, where $\boldsymbol{u}^r$ is the $r$-th column of $\mathbf{U}$ and $\sigma^r$ is the $r$-th diagonal entry of $\boldsymbol{\Sigma}$)
**for** $r = 1$ **to** $R$ **do**
  $\boldsymbol{a}_i^r \leftarrow (\boldsymbol{u}^r)^\top \Delta \mathbf{W}_i$ for all $i$
  $\boldsymbol{a}_{\text{merge}}^r \leftarrow (\boldsymbol{u}^r)^\top \Delta \mathbf{W}_{\text{merge}}$
  $s_i^r \leftarrow \dfrac{\langle \boldsymbol{a}_{\text{merge}}^r, \boldsymbol{a}_i^r \rangle}{\|\boldsymbol{a}_i^r\|_2^2}$ for all $i$
  $\gamma^r \leftarrow K / \sum_{i=1}^{K} \max(\alpha, s_i^r)$
  $\tilde{\sigma}^r \leftarrow \gamma^r \sigma^r$
**end for**
$\Delta \tilde{\mathbf{W}}_{\text{merge}} \leftarrow \mathbf{U}\tilde{\boldsymbol{\Sigma}}\mathbf{V}^\top$
$\mathbf{W}_{\text{merge}} \leftarrow \mathbf{W}_{\text{pre}} + \Delta \tilde{\mathbf{W}}_{\text{merge}}$

---

and output the final merged weights

$$\mathbf{W}_{\text{merge}} = \mathbf{W}_{\text{pre}} + \Delta \tilde{\mathbf{W}}_{\text{merge}}. \qquad (23)$$

## 5. Experiments

### 5.1. Experimental Protocol

**Baselines and Datasets.** We evaluate SVC against representative training-free model merging baselines, including TA (Ilharco et al., 2022), TIES (Yadav et al., 2023), DARE (Yu et al., 2024), TSV-M (Gargiulo et al., 2025), and Iso-CTS (Marczak et al., 2025a). All reported results are produced by our own runs under a unified evaluation protocol. Since the checkpoints used in our study may differ from those in the original papers, absolute numbers can vary from previously reported results. For reference, recent training-free methods often match or exceed training-based approaches such as AdaMerging++ (Yang et al., 2024c) and Surgery (Yang et al., 2024b).

Following common practice (Ilharco et al., 2022; Yang et al., 2024c), we report computer vision (CV) results on 8 multitask classification benchmarks. For natural language processing (NLP), we evaluate on 11 classification benchmarks and additionally report performance on two open LLM leaderboards. Full benchmark lists and evaluation details are provided in the Appendix.

**Implementation Details.** We use the ViT-B/32 CLIP as the default visual encoder, consistent with the setup in (Yang et al., 2024c). The hyperparameters

The hyperparameters of the other compared methods remain identical to those specified in their original papers, and the widely used scaling coefficient $\lambda$ is consistent with 1 after our SVC calibration, thus we remove the variable $\lambda$ in 1.

*Table 1.* **Consolidated average accuracy (%) across CV benchmarks.** Per-dataset results are deferred to the Appendix. Due to the use of different checkpoints, certain methods (*e.g.*, Iso-C and Iso-CTS) differ from those reported in the original paper.

| | 8 Tasks | | | 14 Tasks | | |
|---|---|---|---|---|---|---|
| **Method** | **ViT-B/32** | **ViT-B/16** | **ViT-L/14** | **ViT-B/32** | **ViT-B/16** | **ViT-L/14** |
| Reference (non-merging) | | | | | | |
| Pretrained | 48.0 | 55.2 | 64.9 | 56.6 | 61.7 | 70.4 |
| Individual | 90.5 | 93.0 | 94.4 | 87.3 | 89.5 | 91.4 |
| Training-free merging | | | | | | |
| TA (Ilharco et al., 2022) | 68.9 | 73.7 | 84.3 | 46.4 | 57.1 | 57.7 |
| w/ SVC (Ours) | 81.9 (+13.0 ↑) | 86.2 (+12.5 ↑) | 91.3 (+7.0 ↑) | 63.1 (+16.7 ↑) | 72.0 (+14.9 ↑) | 76.7 (+19.0 ↑) |
| TIES (Yadav et al., 2023) | 72.6 | 76.6 | 85.6 | 61.6 | 60.1 | 62.4 |
| w/ SVC (Ours) | 80.0 (+7.4 ↑) | 84.8 (+8.2 ↑) | 90.6 (+5.0 ↑) | 62.3 (+0.7 ↑) | 63.9 (+3.8 ↑) | 63.6 (+1.2 ↑) |
| DARE (Yu et al., 2024) | 65.8 | 71.5 | 79.4 | 63.9 | 67.0 | 75.4 |
| w/ SVC (Ours) | 80.7 (+14.9 ↑) | 84.8 (+13.3 ↑) | 90.1 (+10.7 ↑) | 71.7 (+7.8 ↑) | 70.0 (+3.0 ↑) | 77.9 (+2.5 ↑) |
| TSV-M (Gargiulo et al., 2025) | 84.0 | 87.3 | 91.5 | 76.3 | 76.6 | 82.3 |
| w/ SVC (Ours) | 84.8 (+0.8 ↑) | 88.0 (+0.7 ↑) | 91.8 (+0.3 ↑) | 76.8 (+0.5 ↑) | 77.0 (+0.4 ↑) | 83.1 (+0.8 ↑) |
| Iso-C (Marczak et al., 2025a) | 83.1 | 87.5 | 91.5 | 73.4 | 69.6 | 76.8 |
| w/ SVC (Ours) | 84.6 (+1.5 ↑) | 88.5 (+1.0 ↑) | 92.2 (+0.7 ↑) | 74.0 (+0.6 ↑) | 71.8 (+2.2 ↑) | 78.5 (+1.7 ↑) |
| Iso-CTS (Marczak et al., 2025a) | 81.4 | 86.9 | 90.9 | 76.7 | 77.6 | 85.7 |
| w/ SVC (Ours) | 85.6 (+4.2 ↑) | 89.7 (+2.8 ↑) | 92.9 (+2.0 ↑) | 76.7 (+0.0 ↑) | 78.5 (+0.9 ↑) | 85.9 (+0.2 ↑) |

*Table 2.* **Consolidated performance across NLP benchmarks.** Details are deferred to the Appendix. Llama2 is evaluated on two generation benchmarks; others are used as encoders for classification.

| | Generative Evaluation | | Encoder-derived Classification | | |
|---|---|---|---|---|---|
| **Method** | **Llama2-7B (FT)** | | **BERT (FT)** | **T5 (FT)** | **T0 (PEFT)** |
| | **AlpacaEval↑** | **GSM8K↑** | **Avg Acc (%)** | **Avg Acc (%)** | **Avg Acc (%)** |
| TA (Ilharco et al., 2022) | 49.1 | 46.1 | 56.9 | 41.5 | 53.5 |
| w/ SVC (Ours) | 51.7 (+2.5 ↑) | 52.2 (+6.1 ↑) | 69.0 (+12.1 ↑) | 46.3 (+4.8 ↑) | 65.8 (+12.3 ↑) |
| TIES (Yadav et al., 2023) | 47.8 | 45.0 | 59.7 | 45.5 | 54.1 |
| w/ SVC (Ours) | 49.2 (+1.3 ↑) | 48.1 (+3.0 ↑) | 61.3 (+1.6 ↑) | 49.7 (+4.2 ↑) | 54.1 (+0.0 ↑) |
| DARE (Yu et al., 2024) | 46.5 | 46.1 | 57.6 | 41.2 | 53.3 |
| w/ SVC (Ours) | 52.8 (+6.3 ↑) | 51.4 (+5.3 ↑) | 57.9 (+0.3 ↑) | 46.2 (+5.0 ↑) | 54.7 (+1.4 ↑) |
| TSV-M (Gargiulo et al., 2025) | 41.7 | 51.9 | 60.6 | 46.5 | — |
| w/ SVC (Ours) | 47.3 (+5.6 ↑) | 51.9 (+0.0 ↑) | 61.3 (+0.8 ↑) | 46.6 (+0.1 ↑) | — |
| Iso-C (Marczak et al., 2025a) | 50.0 | 42.0 | 56.3 | 43.9 | — |
| w/ SVC (Ours) | 58.9 (+8.9 ↑) | 51.4 (+9.4 ↑) | 56.6 (+0.3 ↑) | 48.9 (+5.0 ↑) | — |
| Iso-CTS (Marczak et al., 2025a) | 43.8 | 38.7 | 56.3 | 38.8 | — |
| w/ SVC (Ours) | 51.3 (+7.5 ↑) | 48.0 (+9.3 ↑) | 56.5 (+0.2 ↑) | 46.3 (+7.5 ↑) | — |

Finally, for the only hyperparameter introduced in our paper. Unless otherwise stated, we use $\alpha = 1/K$ following the data-free default, where $K$ is the number of tasks. However, for TSV-M we use $\alpha = 1$, which gives a suppression-only variant and never increases singular values.

### 5.2. Vision & Language: Main Results

**Computer Vision (CV) Experiments**. Following prior work (Marczak et al., 2025a), we evaluate average classification accuracy across 8 and 14 datasets, as shown in Tab. 1 (details are deferred to Appendix C). Our SVC method consistently improves SOTA results in diverse merging tasks.

Notably, without altering the directions of singular vectors, SVC achieves a 19% improvement over Task Arithmetic.

**Natural Language Processing (NLP) Experiments.** We evaluate our method across NLP models of varying sizes in Table 2 (details are deferred to Appendix C). SVC achieves SOTA performance on both conventional small language models and recent large language models (LLMs). For T0, we follow Yu et al. (2024) and adopt $IA^3$-based parameter-efficient fine-tuning (PEFT). Because **$IA^3$ yields task vectors as vectors rather than full weight matrices**, an SVD is not defined; consequently, SVD-dependent approaches (*e.g.*, TSV-M, Iso-C) are not applicable, denoted as —.

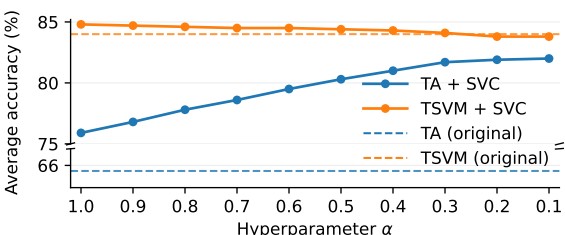

*Figure 5.* **Effect of hyperparameter $\alpha$ in SVC.** When only suppressing over-counting ($\alpha = 1$), SVC yields a stable improvement. In contrast, additionally boosting singular values ($\alpha \in (0, 1)$) requires caution and can degrade performance as $\alpha$ decreases.

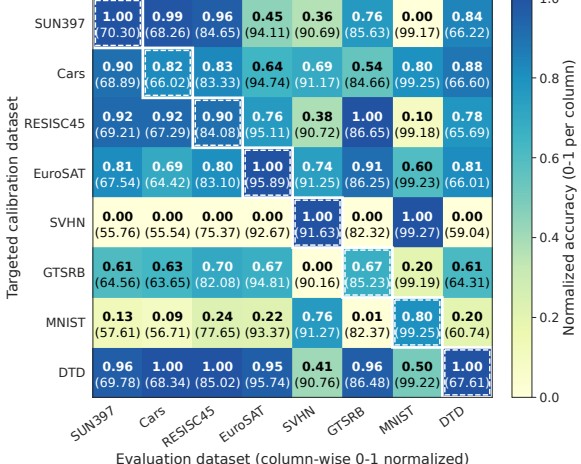

*Figure 6.* **Targeted calibration for a single task.** Each cell $(i, j)$ shows performance on task $j$ after calibrating the merged model using task $i$ as the target. Values are normalized within each column, and the numbers in parentheses show the accuracy.

### 5.3. Empirical Analysis and Ablation Studies

**Ablation Study.** We study the role of the hyperparameter $\alpha$ in SVC. In our calibration rule, $\alpha \in (0, 1]$ controls how aggressively we down-weight over-counted directions by entering the aggregation $\gamma^r = K / \sum_{i=1}^{K} \max(\alpha, s_i^r)$. When $\alpha = 1$, SVC becomes a suppression-only variant and focuses on correcting singular-value inflation caused by spectral over-counting. Smaller $\alpha$ reduces the floor on $\max(\alpha, s_i^r)$ and can yield $\gamma^r > 1$, which amplifies subspaces that are under-accumulated.

Fig. 5 shows that $\alpha = 1$ yields consistent gains, supporting our main contributor. In contrast, allowing additional amplification by decreasing $\alpha$ can produce mixed outcomes, since boosting subspaces may disturb the spectral balance.

**Preference Optimisation.** SVC can favor a specific task after merging. We compute the calibration strength using a chosen target task (referred to as preference optimization). Concretely, we use the target task's $s_i^r$ and set

*Table 3.* **Ablation on singular vector side.** Replacing left singular vectors with right singular vectors for overlap measurement and calibration significantly degrades performance, highlighting the necessity of output-space calibration.

| Method | original | SVC (col, ours) | SVC (row) |
|---|---|---|---|
| TA | 68.9 | 81.9 (+13.0 ↑) | 64.9 (-4.0 ↓) |
| TIES | 72.6 | 80.0 (+7.4 ↑) | 65.7 (-6.9 ↓) |
| DARE | 65.8 | 80.7 (+14.9 ↑) | 67.5 (+1.7 ↑) |
| TSV-M | 84.0 | 84.8 (+0.8 ↑) | 84.0 (+0.0 ↑) |
| Iso-C | 83.1 | 84.6 (+1.5 ↑) | 82.1 (-1.0 ↓) |
| Iso-CTS | 81.4 | 85.6 (+4.2 ↑) | 85.5 (+4.1 ↑) |

*Table 4.* **Runtime and memory footprint overhead of SVC.** SVC applies SVD and runs once offline, without any training.

| Backbone | Time Cost | Memory Usage |
|---|---|---|
| ViT-B/32 | 5.1 s | 1,027.4 MiB |
| ViT-B/16 | 8.2 s | 1,082.8 MiB |
| ViT-L/14 | 15.6 s | 1,488.5 MiB |
| LLaMA2 7B | 517.2 s | 1,898.7 MiB |
| Qwen2.5 7B | 249.3 s | 2,513.1 MiB |

$\gamma^r = 1 / \max(\alpha, s_i^r)$, rather than aggregating across tasks.

Fig. 6 summarizes the effect. Each cell $(i, j)$ calibrates the merge for target task $i$ and evaluates on task $j$. Diagonal entries are usually the largest, indicating that targeting task $i$ primarily improves task $i$. At the same time, tasks that share related features can benefit from the same calibration, while tasks with large domain gaps may degrade, *e.g.*, calibrating for `Cars` improves `SUN397`.

**Output-space vs. Input-space Calibration.** SVC measures subspace-wise overlap in the merged output-space basis, following our projection analysis in Section 3.2. To test whether the choice of singular-vector side matters, we construct a input-space variant that replaces the left singular vectors with the right singular vectors when computing subspace overlap, while keeping all other settings unchanged. Table 3 shows that this input-space variant is far less reliable. It often removes the gains brought by SVC and can even reduce performance below the uncalibrated baseline (for example, `TIES`). This gap is consistent with our analysis. Right singular vectors describe input-side directions, so their overlap reflects how task matrices align with specific input patterns. Overall, the results support output-space overlap as the appropriate quantity for singular-value calibration.

**Cost Analysis.** Consistent with TSV-M and ISO-CTS, our method applies SVD, which introduces additional cost. Even on 7B-parameter LLMs, this overhead is acceptable. Overall, it is far cheaper than training-based methods, because it requires no gradient computation. Table 4 summarizes SVC's runtime and memory across backbones.

# 6. Conclusion

We show that model merging can fail due to spectral over-counting. Using projections onto the merged output-space basis, we find that shared directions are primarily concentrated in top spectral subspaces, which can lead to singular-value inflation. We propose Singular Value Calibration (SVC). SVC measures subspace-wise overlap from these projections and rescales the corresponding singular values, while keeping the spectral directions fixed. Across vision and language benchmarks, SVC consistently improves merging methods and achieves SOTA results.

# Acknowledgements

This work was supported by NSFC Project (62536005, 62192783, 62506162, 624B2063) and Jiangsu Science and Technology Project (BF2025061, BK20251241), Fundamental Research Funds for the Central Universities (KG202508), Postgraduate Research & Practice Innovation Program of Jiangsu Province (KYCX25_0319), Fundamental and Interdisciplinary Disciplines Breakthrough Plan of the Ministry of Education of China (No. JYB2025XDXM118), "111 Center" (No. B26023).

# Impact Statement

This paper presents work whose goal is to advance the field of Machine Learning. There are many potential societal consequences of our work, none which we feel must be specifically highlighted here.

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

# A. Additional Implementation Setting

**Datasets.** Consistent with prior studies (Ilharco et al., 2022; Yang et al., 2024c), our primary experiments are conducted on 8 image classification benchmarks with various domain shift: SUN397 (Xiao et al., 2016), Cars (Krause et al., 2013), RESISC45 (Cheng et al., 2017), EuroSAT (Helber et al., 2019), SVHN (Netzer et al.), GTSRB (Stallkamp et al., 2011), MNIST (MNI), and DTD (Cimpoi et al., 2014). To further demonstrate the versatility of our method, we extend our evaluation to some additional datasets: Caltech101 (Fei-Fei et al., 2004), CIFAR10 (Krizhevsky), CIFAR100 (Krizhevsky), FGVC (Maji et al., 2013), Flowers102 (Nilsback & Zisserman, 2008), Food101 (Bossard et al., 2014), OxfordPets (Parkhi et al., 2012), STL10 (Coates et al., 2011), PCAM (Veeling et al., 2018), FER2013 (Goodfellow et al., 2013), EMNIST (Cohen et al., 2017), FashionMNIST (Xiao et al., 2017), RenderedSST2 (Socher et al., 2013) and KMNIST (Clanuwat et al., 2018). We fine-tune BERT on four binary classification datasets: AG News (Zhang et al., 2015), Rotten Tomatoes (Pang & Lee, 2005), CoLA (Warstadt et al., 2019), and SMS (Almeida et al., 2011). The resulting models are merged into a unified model using model merging techniques and evaluated on each task individually. TA and TIES use default settings, while TSV-M and SVC are applied exclusively to BERT's most critical linear layer ("output.dense.weight"). Finally, to evaluate the performance of the merged model on LLMs, the fine-tuned models WizardMath-7B-V1.0 (Luo et al., 2023) and Llama-2-7b-chat-hf (Touvron et al., 2023) are merged and tested on two benchmarks, AlpacaEval (Li et al., 2023) and GSM8K (Cobbe et al., 2021).

**Datasets License.** Datasets distributed under the MIT License include: SVHN (Netzer et al.), STL10 (Coates et al., 2011), EMNIST (Cohen et al., 2017), FashionMNIST (Xiao et al., 2017), and KMNIST (Clanuwat et al., 2018).

Datasets released under various Creative Commons licenses consist of: EuroSAT (Helber et al., 2019), DTD (Cimpoi et al., 2014), RESISC45 (Cheng et al., 2017), Food101 (Bossard et al., 2014)

The following datasets are available strictly for non-commercial research or academic use, typically under custom or restrictive academic licenses: SUN397 (Xiao et al., 2016), Cars (Krause et al., 2013), GTSRB (Stallkamp et al., 2011), FGVC (Maji et al., 2013), Flowers102 (Nilsback & Zisserman, 2008), OxfordPets (Parkhi et al., 2012), Caltech101 (Fei-Fei et al., 2004), FER2013 (Goodfellow et al., 2013), PCAM (Veeling et al., 2018), and RenderedSST2 (Socher et al., 2013).

The MNIST (MNI) and CIFAR10/100 (Krizhevsky) datasets are provided for unrestricted research use and are considered to be in the public domain or distributed without explicit license restrictions.

For full details regarding dataset licenses and terms of use, please refer to the official web pages or documentation of the respective datasets.

**Implementation Detail.** All experiments are conducted using PyTorch on a single NVIDIA GeForce A800 GPU. Although prior work (Ilharco et al., 2022; Yadav et al., 2023) relies on an additional hyperparameter $\lambda$ to integrate the task update and pre-trained weight, we keep $\lambda = 1$ throughout our experiments. We note that jointly tuning $\lambda$ on top of SVC (*i.e.*, calibrating singular values and then adjusting the overall update scale) could potentially yield further improvements; however, this introduces additional hyperparameter search and is beyond the scope of this work.

# B. Additional Analysis.

**Why Output-Space Rather Than Input-Space?** We use output-space projections because they provide a data-free surrogate for the layer-wise feature behavior of each task. For a task matrix $\Delta \mathbf{W}_i$ and any input feature $\boldsymbol{x}$, the output change along the merged output direction $\boldsymbol{u}^r$ is $(\boldsymbol{u}^r)^\top \Delta \mathbf{W}_i \boldsymbol{x} = \boldsymbol{a}_i^r \boldsymbol{x}$, where $\boldsymbol{a}_i^r = (\boldsymbol{u}^r)^\top \Delta \mathbf{W}_i$. Thus, $\boldsymbol{a}_i^r$ can be viewed as an input-agnostic response function: it describes how task $i$ contributes to the output subspace $\boldsymbol{u}^r$ for all possible inputs, without requiring data. This makes different tasks directly comparable in the same output-space subspace.

In contrast, input-space projection uses right multiplication, $\Delta \mathbf{W}_i \boldsymbol{v}^r = [\boldsymbol{w}_1^\top \boldsymbol{v}^r, \ldots, \boldsymbol{w}_m^\top \boldsymbol{v}^r]^\top$, which only evaluates the task matrix on a single input direction $\boldsymbol{v}^r$. Moreover, $\boldsymbol{v}^r$ is extracted from the merged update and may mix input patterns from multiple tasks, making it less representative for any individual expert. Therefore, input-space measurements are more direction-dependent and less stable for diagnosing task-wise accumulation, whereas output-space responses more directly capture the subspaces where singular-value over-accumulation appears.

**Difference from Iso-CTS and Subspace-Boosting.** SVC is related to recent spectral-space merging methods, but it studies a different failure mode and uses spectral information in a different way. The traditional merging approach aims to

reduce the gap between the merged and expert models. However, our SVC shifts this gap to a more reasonable output-space, and constructs a seemingly similar output-space matching objective:

$$\min_{\Delta W_{\mathrm{merge}}} \sum_{i=1}^{K} \left\| U_{\mathrm{merge}}^\top \Delta W_{\mathrm{merge}} - U_{\mathrm{merge}}^\top \Delta W_i \right\|_F^2 , \tag{24}$$

where $U_{\mathrm{merge}}$ denotes the output-space basis of the merged update. This objective asks whether the projected merged update is close to each projected expert update. However, in the actual optimization, we slightly modify the above objective. Since our goal is to preserve the behavior of the original expert models as much as possible, we instead minimize the distance between the projection of $U_{\mathrm{merge}}^\top \Delta W_{\mathrm{merge}}$ onto $U_{\mathrm{merge}}^\top \Delta W_i$ and $U_{\mathrm{merge}}^\top \Delta W_i$ itself.

This is fundamentally different from the optimization objectives and operations of other related methods. Iso-C and Iso-CTS study spectral alignment and adjust the merged spectrum to reduce the dominance of large singular components. Their motivation is mainly based on the imbalance between dominant and non-dominant spectral directions. In contrast, SVC explains why such dominance appears from the perspective of task interactions: when multiple tasks contribute positively to the same output subspace, the shared component is repeatedly accumulated, which inflates the corresponding singular value. Thus, SVC provides a task-wise and subspace-wise calibration rule rather than a global spectral correction.

SVC is also different from Subspace-Boosting. Subspace-Boosting focuses on rank collapse and underutilized spectral dimensions, and therefore aims to recover or boost missing subspace capacity. SVC instead targets the opposite but complementary phenomenon: dominant shared subspaces can be over-accumulated after merging. Rather than boosting collapsed directions or constructing a new merged update, SVC is a post-hoc calibration method. Given an already merged update, it keeps the singular vectors fixed and only rescales singular values according to the measured over-counting coefficients. Therefore, SVC can be used as a plug-in correction on top of existing merging methods, including Iso-style or other spectral-space methods.

**Broader Context: Alignment, Redundancy, and Compact Structure.** Our analysis suggests that the challenge in model merging is not merely the lack of shared structure, but the redundant accumulation of aligned components in a few dominant output subspaces. This perspective is connected to broader studies on representation alignment and redundancy. For example, subspace- or kernel-based measures such as SVCCA and CKA have been used to compare neural representations across models or training stages (Raghu et al., 2017; Kornblith et al., 2019). Related redundancy-oriented studies aim to reduce redundant feature dimensions in self-supervised representations (Zbontar et al., 2021), identify compact subnetworks that preserve model capability under the lottery-ticket view (Frankle & Carbin, 2018; Liu et al., 2024; Jiang et al., 2023), or characterize informational uniqueness for efficient video compression (Yuan et al., 2025a). Recent works also study how to separate stable or identity-preserving information from redundant or nuisance factors in retrieval and person re-identification (Yuan et al., 2025c;d;b). Similar alignment problems further arise when heterogeneous sources must be compared in a shared space, such as cross-subject fMRI decoding, where Duala aligns subject-level and stimulus-level factors (Li et al., 2026). Different from these representation-level or sparsity-oriented studies, SVC operates directly on merged weight updates: it uses the merged output-space basis as a common coordinate system, measures task-wise over-accumulation within each spectral subspace, and corrects only the corresponding singular values. These works therefore serve as broader context for the shared principle that effective transfer should preserve useful common structure while controlling harmful redundancy or misalignment.

## C. Additional Experiments.

Here, we present experiments that were not included in the main paper due to space limitations.

**More Number of Merged Tasks.** We further evaluate SVC in a more challenging 20-task CV merging setting. As the number of merged tasks increases, preserving all task-specific knowledge becomes harder because more shared and task-specific components are mixed within the same merged spectral basis. As shown in Table 5, SVC still brings consistent improvements over different base merging methods, improving Subspace-Boosted from 68.89% to 71.47%, TALL Mask from 37.94% to 54.41%, TSV-M from 76.63% to 76.98%, Iso-C from 75.21% to 75.90%, and Iso-CTS from 77.64% to 77.72%. These results indicate that spectral over-accumulation also appears when more models are merged, and SVC remains an effective post-hoc calibration module under larger-scale merging settings.

*Table 5.* Average accuracy (%) on the 20-task CV merging setting. SVC is applied as a post-hoc calibration on top of each base merger.

| Method | Task Arithmetic | Subspace-Boosted | TALL Mask | TSV-M | Iso-C | Iso-CTS |
|---|---|---|---|---|---|---|
| w/o SVC | 60.98 | 68.89 | 37.94 | 76.63 | 75.21 | 77.64 |
| w/ SVC | 61.19 | 71.47 | 54.41 | 76.98 | 75.90 | 77.72 |
| Gain | +0.21 | +2.58 | +16.47 | +0.35 | +0.69 | +0.08 |

*Table 6.* End-to-end runtime of base merge operators with and without SVC. SVC is executed once offline and adds limited overhead.

| Method | ViT-B/32 (8 Tasks) | LLaMA2-7B | BERT |
|---|---|---|---|
| TA | 0.2s | 52.4s | 0.1s |
| TA w/ SVC | 4.7s | 532.3s | 6.1s |
| TIES | 9.5s | 156.6s | 3.9s |
| TIES w/ SVC | 13.4s | 656.6s | 10.0s |
| DARE | 12.5s | 184.8s | 4.2s |
| DARE w/ SVC | 16.3s | 698.7s | 9.5s |
| TSV-M | 28.5s | 238.1s | 20.9s |
| TSV-M w/ SVC | 32.4s | 750.4s | 25.9s |
| Iso-C | 4.2s | 269.0s | 4.6s |
| Iso-C w/ SVC | 9.6s | 769.2s | 13.7s |
| Iso-CTS | 123.6s | 378.7s | 75.1s |
| Iso-CTS w/ SVC | 128.5s | 882.5s | 82.8s |

**Overall Computational Cost with Base Merge Operators.** We further report the end-to-end cost of applying SVC on top of different base merge operators:

$$T_{\text{total}} = T_{\text{base}} + T_{\text{SVC}}, \tag{25}$$

where $T_{\text{base}}$ is the runtime of the base merger and $T_{\text{SVC}}$ is the additional post-hoc calibration cost. As shown in Table 6, SVC only introduces a moderate offline overhead. On ViT-B/32, it adds only several seconds for most base operators. On LLaMA2-7B, the cost is higher due to SVD over larger matrices, but the total runtime remains practical and is paid only once after merging. Since SVC requires no gradients, no auxiliary data, and no iterative optimization, it introduces no inference-time overhead and remains lightweight compared with training or fine-tuning.

**Weight Averaging and Spectral Imbalance.** Even under weight averaging (WA) $\Delta\mathbf{W}_{\text{merge}} = \frac{1}{K} \sum_{i=1}^{K} \Delta\mathbf{W}_i$. Fig. 3 indicates that, in the first few subspaces, the original singular value can be much larger than $K \cdot \sigma_i^\star$ (with $K$ tasks), while the $1/K$ factor can make many other singular values overly small.

**Merging Experiments on 8 CV Benchmark.** The performance of each method on individual datasets is presented in detail. Complete results for ViT-B/32, ViT-B/16, and ViT-L/14 are provided in Table 7, Table 8, and Table 9, respectively. We additionally compare with CAT Merging (Sun et al., 2025a) and STAR (Lee et al., 2025). Due to limited time and computing resources, we did not re-run these baselines across all backbones and settings; instead, we report results only on the canonical ViT-B/32 8-task benchmark. As shown in Table 7, both CAT and STAR are substantially lower than our SVC-enhanced method.

**Merging Experiments on 14 CV Benchmark.** The performance of each method on individual datasets is presented in detail. Complete results for ViT-B/32, ViT-B/16, and ViT-L/14 are provided in Table 10, Table 11, and Table 12, respectively.

**Merging Experiments on Full Fine-Tuned BERT.** The performance of each method on individual datasets is presented in detail. Complete results for BERT (Devlin et al., 2019) are provided in Table 13.

**Merging Experiments on Full Fine-Tuned T5.** The performance of each method on individual datasets is presented in detail. Complete results for T5 (Raffel et al., 2020) are provided in Table 14.

*Table 7.* Comparison of different model merging methods across eight vision benchmarks on ViT-B/32. Bold values indicate the best performance among merging-based techniques. The notation "(Ours)" highlights the integration of our proposed SVC method.

| METHOD | SUN397 | CARS | RESISC45 | EuroSAT | SVHN | GTSRB | MNIST | DTD | **AVG.** |
|---|---|---|---|---|---|---|---|---|---|
| PRETRAINED | 62.3 | 59.7 | 60.7 | 45.5 | 31.4 | 32.6 | 48.5 | 43.8 | 48.1 |
| INDIVIDUAL | 75.3 | 77.7 | 96.1 | 99.7 | 97.5 | 98.7 | 99.7 | 79.4 | 90.5 |
| TRADITIONAL MTL | 73.9 | 74.4 | 93.9 | 98.2 | 95.8 | 98.9 | 99.5 | 77.9 | 89.1 |
| TA | 55.1 | 54.9 | 66.7 | 77.2 | 80.2 | 69.7 | 97.3 | 50.1 | 68.9 |
| W/ SVC (OURS) | 68.1 | 66.3 | 83.5 | 95.6 | 91.2 | 86.0 | 99.2 | 65.6 | 81.9 (+13.0 ↑) |
| DARE | 64.8 | 63.5 | 71.8 | 72.4 | 63.8 | 52.4 | 87.5 | 50.5 | 65.8 |
| W/ SVC (OURS) | 68.0 | 66.6 | 82.8 | 92.9 | 88.8 | 84.4 | 99.1 | 62.9 | 80.7 (+14.9 ↑) |
| TIES | 59.6 | 58.6 | 71.0 | 81.3 | 86.1 | 70.9 | 98.4 | 54.7 | 72.6 |
| W/ SVC (OURS) | 69.0 | 66.0 | 83.0 | 90.8 | 88.7 | 78.6 | 98.7 | 65.0 | 80.0 (+7.4 ↑) |
| TSV-M | 69.1 | 70.7 | 85.5 | 94.3 | 92.0 | 91.9 | **99.3** | 69.2 | 84.0 |
| W/ SVC (OURS) | 70.3 | 72.4 | 86.5 | 94.9 | **92.1** | **92.4** | **99.3** | 70.0 | 84.8 (+0.8 ↑) |
| ISO-C | 74.8 | 74.1 | 87.9 | 92.9 | 83.1 | 86.0 | 98.2 | 67.9 | 83.1 |
| W/ SVC (OURS) | 74.1 | 72.6 | 88.1 | **95.3** | 88.2 | 89.6 | 99.0 | 70.0 | 84.6 (+1.5 ↑) |
| ISO-CTS | 74.4 | 74.4 | 87.2 | 90.4 | 76.8 | 83.3 | 97.4 | 67.0 | 81.4 |
| W/ **SVC** | **75.6** | **75.2** | **90.1** | 94.9 | 86.1 | 91.6 | 98.9 | **72.1** | **85.6** (+4.2 ↑) |
| STAR | 55.9 | 55.1 | 67.4 | 77.7 | 80.2 | 68.1 | 97.2 | 50.1 | 69.0 |
| CAT | 68.1 | 65.4 | 80.5 | 89.5 | 85.5 | 78.5 | 98.6 | 60.7 | 78.4 |

**Merging Experiments on PEFT Fine-Tuned T0.** We report detailed performance on each dataset and summarize the full results for T0 (Sanh et al., 2021) in Table 15. We adopt IA$^3$ (Liu et al., 2022) for parameter-efficient fine-tuning. Across datasets, adding our SVC method consistently improves merged performance, highlighting the benefit of calibrating over-accumulated magnitudes during merging. Notably, methods that rely on SVD (*e.g.*, TSV-M, Iso-C, Iso-CTS) are not applicable in this setting because IA$^3$ updates are one-dimensional vectors rather than full weight matrices.

To make SVC applicable to 1D updates, we derive a vector-form calibration that keeps the merged direction unchanged and only corrects its scale. For each layer, let ($\{\boldsymbol{\tau}_i\}_{i=1}^K$) be the task vectors from (K) experts and let ($\boldsymbol{\tau}_{\mathrm{merge}}$) be an initial merged vector (we use simple averaging). We measure how much the merged vector can be explained by each expert via a projection coefficient

$$s_i = \frac{\langle \boldsymbol{\tau}_{\mathrm{merge}}, \boldsymbol{\tau}_i \rangle}{\langle \boldsymbol{\tau}_i, \boldsymbol{\tau}_i \rangle}, \tag{26}$$

and aggregate them by ($\gamma = \frac{K}{\sum_{i=1}^K s_i}$). A smaller ($\gamma$) indicates that the merged update is over-counted along expert directions, which is the 1D counterpart of singular-value accumulation. We then rescale the merged vector as ($\tilde{\boldsymbol{\tau}}_{\mathrm{merge}} = \gamma \boldsymbol{\tau}_{\mathrm{merge}}$). This calibration is lightweight, requires no data, and extends SVC to PEFT scenarios where SVD-based baselines cannot operate.

**Generality Beyond Matrix SVD.** In practice, we apply SVC in a layer-wise manner: we perform SVD and calibration separately for the weight update matrix of each linear layer, and then reconstruct the calibrated update for that layer.

While our primary implementation is described for 2D weight matrices (where a standard SVD is directly applicable), the underlying principle of SVC, **measuring over-accumulation along shared directions and calibrating the magnitude without changing the direction**, extends naturally to other parameter shapes. For higher-order weight tensors, one can apply the same idea after tensor unfolding/reshaping into a matrix (or other appropriate spectral decomposition); for 1D parameter updates, SVD is undefined but the same "calibrate-the-scale" rule can be derived using vector projections. Our T0 (PEFT/IA$^3$) experiments provide a concrete example of this extension, where IA$^3$ produces one-dimensional updates and SVC still yields consistent gains.

**Positioning vs. Existing Spectral-Domain Merging.** SVC also operates in spectral space, but it serves a different purpose from prior SVD-based baselines and can be used as a drop-in post hoc calibration on top of an existing merge. Methods such as TSV-M focus on constructing a merged update that is less affected by conflicts, for example by selecting or reweighting directions to reduce destructive interference. The Iso-* family highlights that dominant singular components can suppress smaller ones, which is an important observation. However, in a single model the top singular values are naturally much larger

*Table 8.* Comparison of different model merging methods across eight vision benchmarks on ViT-B/16. Bold values indicate the best performance among merging-based techniques. The notation "(Ours)" highlights the integration of our proposed SVC method.

| METHOD | SUN397. | CARS. | RESISC45. | EUROSAT. | SVHN. | GTSRB. | MNIST. | DTD. | **AVG.** |
|---|---|---|---|---|---|---|---|---|---|
| PRETRAINED | 63.8 | 64.7 | 66.4 | 54.6 | 52.0 | 43.4 | 51.7 | 44.7 | 55.2 |
| INDIVIDUAL | 81.8 | 86.8 | 96.9 | 99.8 | 97.9 | 99.2 | 99.8 | 82.1 | 93.0 |
| TA | 61.2 | 66.0 | 74.5 | 74.4 | 88.1 | 73.9 | 98.5 | 52.7 | 73.7 |
| W/ SVC (OURS) | 72.8 | 77.5 | 87.6 | 96.5 | 93.1 | 91.9 | 99.2 | 71.1 | 86.2 (+12.5 ↑) |
| DARE | 67.6 | 70.0 | 76.0 | 78.6 | 75.3 | 59.8 | 94.4 | 50.1 | 71.5 |
| W/ SVC (OURS) | 72.6 | 77.4 | 86.6 | 95.2 | 91.7 | 88.9 | 99.1 | 67.3 | 84.8 (+13.3 ↑) |
| TIES | 66.4 | 70.5 | 79.8 | 80.4 | 89.9 | 70.3 | 98.8 | 57.1 | 76.6 |
| W/ SVC (OURS) | 75.0 | 76.8 | 88.5 | 94.8 | 91.2 | 82.7 | 99.0 | 70.7 | 84.8 (+8.2 ↑) |
| TSV-M | 72.8 | 80.3 | 89.1 | 96.6 | **93.9** | 94.0 | **99.3** | 72.7 | 87.3 |
| W/ SVC (OURS) | 73.9 | 81.3 | 89.8 | 97.3 | 93.8 | 94.8 | **99.3** | 73.7 | 88.0 (+0.7 ↑) |
| ISO-C | 78.1 | 82.3 | 91.9 | 96.9 | 88.3 | 91.8 | 98.8 | 71.9 | 87.5 |
| W/ SVC (OURS) | 77.5 | 81.8 | 92.0 | 97.5 | 91.6 | 94.4 | 99.1 | 74.1 | 88.5 (+1.0 ↑) |
| ISO-CTS | 77.9 | 83.2 | 92.0 | 96.4 | 84.9 | 91.3 | 98.4 | 71.1 | 86.9 |
| W/ SVC (OURS) | **78.6** | **83.7** | **93.6** | **98.0** | 90.5 | **96.6** | 99.1 | **77.0** | **89.7** (+2.8 ↑) |
| STAR | 63.2 | 66.3 | 73.7 | 79.0 | 85.6 | 76.4 | 98.4 | 51.8 | 74.3 |
| CAT | 72.9 | 75.9 | 83.1 | 92.8 | 88.2 | 82.7 | 98.8 | 62.7 | 82.1 |

than the tail of the spectrum. What is missing is a task-interaction explanation that attributes this suppression to repeated accumulation of shared directions across tasks, rather than to inherently larger singular values in the leading subspaces.

Building on our analysis, SVC targets this specific failure mode. When multiple tasks align in the same subspaces, naive aggregation can repeatedly add the same shared components, inflate the subspace strength, and concentrate the merged spectrum. SVC quantifies the degree of this over-counting in each subspace using projection-based coefficients, then corrects the magnitude while keeping the spectral directions unchanged. As a result, **SVC is complementary to strong spectral baselines.** When conflicts have already been largely mitigated, the remaining room for improvement can be small, which explains the typically modest but consistent gains on TSV-M. At the same time, methods that do not explicitly control subspace magnitudes tend to benefit more from this calibration.

*Table 9.* Comparison of different model merging methods across eight vision benchmarks on ViT-L/14. Bold values indicate the best performance among merging-based techniques. The notation "(Ours)" highlights the integration of our proposed SVC method.

| Method | SUN397. | Cars. | RESISC45. | EuroSAT. | SVHN. | GTSRB. | MNIST. | DTD. | Avg. |
|---|---|---|---|---|---|---|---|---|---|
| Pretrained | 66.9 | 77.9 | 71.3 | 62.2 | 58.5 | 50.6 | 76.4 | 55.4 | 64.9 |
| Individual | 84.9 | 92.4 | 97.4 | 99.7 | 98.1 | 99.2 | 99.7 | 84.2 | 94.4 |
| TA | 73.9 | 82.1 | 86.7 | 92.7 | 87.9 | 86.8 | 98.9 | 65.6 | 84.3 |
| w/ SVC (Ours) | 80.9 | 89.5 | 93.2 | 98.6 | 93.8 | 96.3 | 99.4 | 78.8 | 91.3 (+7.0 ↑) |
| DARE | 71.1 | 81.6 | 82.6 | 90.6 | 78.3 | 70.8 | 97.0 | 63.1 | 79.4 |
| w/ SVC (Ours) | 79.3 | 88.1 | 92.6 | 97.7 | 92.5 | 94.8 | 99.3 | 76.4 | 90.1 (+10.7 ↑) |
| TIES | 76.4 | 84.2 | 88.9 | 95.2 | 90.0 | 83.0 | 99.0 | 67.9 | 85.6 |
| w/ SVC (Ours) | 81.7 | 89.4 | 93.7 | 98.1 | 92.7 | 92.0 | 99.3 | 78.1 | 90.6 (+5.0 ↑) |
| TSV-M | 79.0 | 89.8 | 94.0 | 98.8 | 95.3 | 96.2 | **99.5** | 79.1 | 91.5 |
| w/ SVC (Ours) | 79.4 | 90.3 | 94.2 | **98.9** | **95.6** | 96.8 | **99.5** | 79.9 | 91.8 (+0.3 ↑) |
| ISO-C | 81.9 | 90.9 | 94.8 | 98.7 | 91.4 | 95.5 | 99.2 | 79.2 | 91.5 |
| w/ SVC (Ours) | 82.7 | 90.6 | 94.8 | 98.5 | 93.7 | 96.7 | 99.4 | 80.8 | 92.2 (+0.7 ↑) |
| ISO-CTS | 81.3 | 91.2 | 94.7 | 98.6 | 89.3 | 95.3 | 99.2 | 77.9 | 90.9 |
| w/ SVC (Ours) | **83.3** | **91.9** | **96.0** | 98.8 | 93.8 | **97.8** | 99.4 | **82.4** | **92.9** (+2.0 ↑) |
| STAR | 74.5 | 82.0 | 86.7 | 93.1 | 87.8 | 87.3 | 98.8 | 65.0 | 84.4 |
| CAT | 78.7 | 88.5 | 91.1 | 96.3 | 91.3 | 95.7 | 99.4 | 75.7 | 89.6 |

*Table 10.* Comparison of model merging methods across 14 benchmarks on ViT-B/32. Bold values indicate the best performance among merging-based techniques. The notation "(Ours)" highlights the integration of our proposed SVC method.

| Method | Cal101 | Cars | CIF100 | DTD | Euro | FGVC | Flo102 | Food | GTSRB | MNIST | OxfP | RESISC | SUN | SVHN | Avg. |
|---|---|---|---|---|---|---|---|---|---|---|---|---|---|---|---|
| Pretrained | 89.2 | 59.6 | 66.1 | 44.4 | 45.7 | 17.0 | 73.5 | 79.5 | 32.6 | 48.3 | 82.3 | 60.3 | 62.3 | 31.6 | 56.6 |
| Individual | 95.1 | 77.7 | 89.3 | 79.4 | 99.8 | 46.6 | 87.3 | 85.0 | 98.7 | 99.7 | 90.5 | 96.1 | 79.2 | 97.5 | 87.3 |
| TA | 58.2 | 36.7 | 46.3 | 32.7 | 61.6 | 17.4 | 29.4 | 35.6 | 49.6 | 91.4 | 62.8 | 42.0 | 27.1 | 58.8 | 46.4 |
| w/ SVC (Ours) | 87.8 | 53.2 | 64.7 | 53.5 | 73.7 | 30.3 | 47.1 | 53.8 | 66.7 | 95.1 | 73.6 | 67.0 | 57.4 | 60.0 | 63.1 (+16.7 ↑) |
| DARE | **92.7** | 62.4 | 71.7 | 46.7 | 64.3 | 18.5 | **73.5** | **78.9** | 43.7 | 76.7 | 85.0 | 67.0 | 63.6 | 51.4 | 63.9 |
| w/ SVC (Ours) | **92.7** | 63.6 | 76.7 | 55.9 | 85.3 | 30.2 | 64.7 | 76.9 | 67.5 | 94.1 | 85.3 | 74.9 | 66.2 | 70.6 | 71.7 (+7.8 ↑) |
| TIES | 86.6 | 55.9 | 69.7 | 47.0 | 70.6 | 30.0 | 49.0 | 65.1 | 53.0 | 87.7 | 69.8 | 63.5 | 59.0 | 55.1 | 61.6 |
| w/ SVC (Ours) | 89.2 | 55.6 | 70.8 | 49.5 | 68.7 | 33.9 | 49.0 | 65.7 | 52.4 | 87.4 | 70.1 | 64.7 | 62.2 | 52.8 | 62.3 (+0.7 ↑) |
| TSV-M | 90.6 | 67.1 | 74.2 | 65.6 | **94.6** | 37.1 | 59.8 | 73.9 | **88.4** | 99.0 | 86.7 | 81.1 | 64.1 | **86.9** | 76.3 |
| w/ SVC (Ours) | 91.8 | 67.2 | 74.3 | 65.7 | 94.3 | 38.1 | 61.8 | 74.3 | **88.4** | 99.0 | 86.7 | 81.3 | 65.2 | 86.6 | 76.8 (+0.5 ↑) |
| ISO-C | 91.2 | 63.9 | 75.5 | 61.2 | 90.9 | 39.0 | 50.0 | 69.2 | 83.6 | 98.5 | 78.8 | 77.5 | 65.5 | 82.7 | 73.4 |
| w/ SVC (Ours) | 91.5 | 65.2 | 75.7 | 62.3 | 91.4 | 39.2 | 54.9 | 70.3 | 82.3 | 98.3 | 78.8 | 78.2 | 66.7 | 80.8 | 74.0 (+0.6 ↑) |
| ISO-CTS | **92.7** | **68.7** | 77.1 | 64.9 | 89.8 | 39.2 | 64.7 | 75.5 | 85.8 | 98.5 | 83.4 | **82.8** | 68.7 | 82.4 | 76.7 |
| w/ SVC (Ours) | 91.3 | 68.0 | **78.1** | **66.3** | 91.0 | **39.8** | 63.7 | 75.1 | 85.0 | 98.5 | 82.9 | **84.0** | **69.4** | 80.5 | 76.7 (+0.0 ↑) |

*Table 11.* Comparison of model merging methods across 14 benchmarks on ViT-B/16. Bold values indicate the best performance among merging-based techniques. The notation "(Ours)" highlights the integration of our proposed SVC method.

| Method | Cal101 | Cars | CIF100 | DTD | Euro | FGVC | Flo102 | Food | GTSRB | MNIST | OxfP | RESISC | SUN | SVHN | Avg. |
|---|---|---|---|---|---|---|---|---|---|---|---|---|---|---|---|
| Pretrained | 86.7 | 64.7 | 69.6 | 44.7 | 54.6 | 25.1 | 67.7 | 85.7 | 43.4 | 51.7 | 87.2 | 66.4 | 63.8 | 52.0 | 61.7 |
| Individual | 95.7 | 86.8 | 83.0 | 82.2 | 99.8 | 46.1 | 82.4 | 88.9 | 99.2 | 99.8 | 94.6 | 96.9 | 82.0 | 97.9 | 88.2 |
| TA | 90.1 | 43.5 | 66.7 | 36.4 | 54.7 | 20.3 | 47.1 | 61.2 | 47.9 | 86.0 | 82.1 | 52.0 | 53.5 | 57.9 | 57.1 |
| w/ SVC (Ours) | 95.0 | 57.6 | 77.7 | 48.0 | 80.4 | 33.5 | 65.7 | 77.4 | 71.3 | 97.7 | 90.5 | 71.9 | 63.1 | 77.9 | 72.0 (+14.9 ↑) |
| DARE | 88.2 | 66.4 | 76.2 | 46.4 | 65.5 | 27.0 | 70.6 | **84.3** | 51.0 | 78.7 | 86.7 | 69.8 | 65.6 | 62.3 | 67.0 |
| w/ SVC (Ours) | 93.5 | 56.2 | 78.6 | 45.1 | 76.5 | 32.0 | 64.7 | 78.4 | 65.8 | 96.0 | 87.5 | 68.6 | 62.4 | 74.4 | 70.0 (+3.0 ↑) |
| TIES | 90.6 | 51.4 | 81.6 | 38.9 | 47.6 | 28.4 | 57.8 | 76.9 | 41.0 | 78.8 | 84.2 | 52.1 | 60.3 | 51.2 | 60.1 |
| w/ SVC (Ours) | 89.8 | 56.2 | **82.8** | 42.4 | 56.0 | 31.2 | 63.7 | 82.9 | 45.0 | 78.9 | 87.8 | 59.1 | 62.9 | 56.0 | 63.9 (+3.8 ↑) |
| TSV-M | 91.9 | 67.0 | 79.9 | 53.0 | 90.0 | 37.7 | 71.6 | 81.9 | 82.8 | 98.4 | 93.2 | 77.4 | 64.9 | 82.7 | 76.6 |
| w/ SVC (Ours) | 92.1 | 67.1 | 79.9 | 53.3 | 90.4 | **38.6** | 73.5 | 82.3 | 83.5 | 98.4 | **93.5** | 78.0 | 65.1 | 83.0 | 77.0 (+0.4 ↑) |
| ISO-C | **95.5** | 44.4 | 80.0 | 42.5 | 79.5 | 37.4 | 57.8 | 75.1 | 74.0 | 97.7 | 90.5 | 64.8 | 56.8 | 78.0 | 69.6 |
| w/ SVC (Ours) | 95.1 | 57.1 | 79.9 | 46.8 | 80.6 | 36.0 | 63.7 | 78.6 | 71.8 | 97.2 | 90.0 | 70.5 | 62.5 | 75.6 | 71.8 (+2.2 ↑) |
| ISO-CTS | 95.0 | 66.5 | 76.4 | 54.0 | **92.6** | 38.3 | 74.5 | 80.3 | **88.9** | 98.8 | 92.7 | 78.2 | 63.4 | **87.3** | 77.6 |
| w/ SVC (Ours) | 94.7 | **70.2** | 77.8 | **55.7** | 92.4 | 38.4 | **77.5** | 82.8 | 85.3 | 98.5 | 92.9 | **81.2** | 67.3 | 84.9 | **78.5** (+0.9 ↑) |

*Table 12.* Comparison of model merging methods across 14 benchmarks on ViT-L/14. Bold values indicate the best performance among merging-based techniques. The notation "(Ours)" highlights the integration of our proposed SVC method.

| METHOD | CAL101 | CARS | CIF100 | DTD | EURO | FGVC | FLO102 | FOOD | GTSRB | MNIST | OXFP | RESISC | SUN | SVHN | AVG. |
|---|---|---|---|---|---|---|---|---|---|---|---|---|---|---|---|
| PRETRAINED | 91.4 | 77.9 | 78.5 | 55.4 | 62.3 | 31.5 | 81.4 | 89.6 | 50.5 | 76.3 | 93.8 | 71.3 | 66.9 | 58.4 | 70.4 |
| INDIVIDUAL | 95.8 | 92.3 | 87.8 | 84.1 | 99.7 | 65.0 | 88.2 | 92.6 | 99.2 | 99.7 | 94.3 | 97.4 | 84.9 | 98.1 | 91.4 |
| TA | 91.9 | 36.0 | 78.5 | 41.4 | 52.6 | 25.2 | 60.8 | 56.0 | 46.6 | 84.2 | 85.9 | 48.3 | 55.8 | 45.1 | 57.7 |
| W/ SVC (OURS) | 93.7 | 66.8 | 86.4 | 56.9 | 81.2 | 48.5 | 74.5 | 79.1 | 76.6 | 97.8 | 92.1 | 74.4 | 66.4 | 78.9 | **76.7** (+19.0 ↑) |
| DARE | 92.5 | 76.9 | 85.2 | 58.0 | 78.5 | 33.9 | 81.4 | **88.5** | 62.2 | 91.5 | 93.8 | 76.6 | 69.0 | 68.1 | 75.4 |
| W/ SVC (OURS) | 93.2 | 70.0 | 86.8 | 56.6 | 83.9 | 47.9 | 76.5 | 82.3 | 77.2 | 97.1 | 93.8 | 77.8 | 68.1 | 79.0 | 77.9 (+2.5 ↑) |
| TIES | 92.5 | 51.1 | 88.7 | 47.9 | 48.9 | 38.1 | 66.7 | 76.1 | 45.9 | 65.7 | 91.3 | 58.5 | 62.3 | 39.5 | 62.4 |
| W/ SVC (OURS) | 92.9 | 51.9 | 88.8 | 46.0 | 49.0 | 48.6 | 61.8 | 77.6 | 45.9 | 66.2 | 93.8 | 60.5 | 61.3 | 46.9 | 63.6 (+1.2 ↑) |
| TSV-M | 94.0 | 77.0 | **88.9** | 62.8 | 92.8 | 51.6 | 80.4 | 85.2 | 89.2 | 98.8 | **94.8** | 83.3 | 69.6 | 84.0 | 82.3 |
| W/ SVC (OURS) | 94.1 | 78.2 | 88.6 | 63.6 | 94.0 | 52.7 | 82.3 | 86.2 | 90.1 | 98.9 | 94.6 | 84.5 | 70.2 | 85.2 | 83.1 (+0.8 ↑) |
| ISO-C | 94.0 | 60.7 | 87.4 | 54.7 | 84.0 | 53.3 | 77.5 | 75.4 | 80.1 | 98.3 | 93.5 | 71.2 | 63.2 | 81.4 | 76.8 |
| W/ SVC (OURS) | 93.8 | 71.7 | 86.8 | 58.1 | 84.7 | 50.0 | 75.5 | 81.9 | 79.4 | 97.9 | 93.5 | 78.3 | 68.8 | 78.5 | 78.5 (+1.7 ↑) |
| ISO-CTS | **94.2** | 82.2 | 87.4 | **70.8** | **96.7** | **57.5** | **90.2** | 85.1 | **94.8** | **99.2** | **94.8** | 84.8 | 71.3 | **90.7** | 85.7 |
| W/ SVC (OURS) | 94.0 | **83.9** | 88.1 | 70.1 | 96.6 | 56.0 | 89.2 | 87.8 | 93.3 | **99.2** | 94.6 | **87.6** | **73.7** | 89.0 | **85.9** (+0.2 ↑) |

*Table 13.* Comparison of model merging methods across three NLP benchmarks on BERT. Bold values indicate the best performance among merging-based techniques. The notation "(Ours)" highlights the integration of our proposed SVC method.

| METHOD | AG_NEWS | ROTTEN_TOMATOES | COLA | AVG. ACC. |
|---|---|---|---|---|
| EXPERT | 99.1 | 84.1 | 78.3 | 87.2 |
| WA | 48.5 | 58.9 | 72.1 | 59.8 |
| TA | 50.4 | 51.1 | 69.1 | 56.9 |
| W/ SVC (OURS) | **67.9** | **78.3** | 60.8 | **69.0** (+12.1 ↑) |
| TIES | 51.6 | 57.4 | 70.1 | 59.7 |
| W/ SVC (OURS) | 52.7 | 61.1 | 70.2 | 61.3 (+1.6 ↑) |
| DARE | 50.0 | 50.4 | 72.4 | 57.6 |
| W/ SVC (OURS) | 50.0 | 50.7 | **73.1** | 57.9 (+0.3 ↑) |
| TSV-M | 59.4 | 58.2 | 64.1 | 60.6 |
| W/ SVC (OURS) | 59.5 | 59.0 | 65.5 | 61.3 (+0.8 ↑) |
| ISO-C | 49.7 | 50.0 | 69.1 | 56.3 |
| W/ SVC (OURS) | 50.4 | 50.2 | 69.1 | 56.6 (+0.3 ↑) |
| ISO-CTS | 49.8 | 50.0 | 69.1 | 56.3 |
| W/ SVC (OURS) | 50.2 | 50.1 | 69.1 | 56.5 (+0.2 ↑) |

*Table 14.* Comparison of model merging methods across 11 NLP benchmarks using bigscience/T5. Bold values indicate the best performance among merging-based techniques. The notation "(Ours)" highlights the integration of our proposed SVC method.

| METHOD | RTE | CB | WINOGR. | WIC | WSC | COPA | H-SWAG | STORY | ANLI-R1 | ANLI-R2 | ANLI-R3 | AVG. |
|---|---|---|---|---|---|---|---|---|---|---|---|---|
| TA | 40.6 | 53.1 | 34.4 | 60.9 | 37.5 | 48.4 | 21.9 | 50.0 | 34.4 | 40.6 | 34.4 | 41.5 |
| W/ SVC (OURS) | 43.8 | 62.5 | 46.9 | 68.8 | 37.5 | 59.4 | 21.9 | 59.4 | 34.4 | **43.8** | 31.2 | 46.3 (+4.8 ↑) |
| DARE | 40.6 | 53.1 | 31.2 | 59.4 | 37.5 | 50.0 | 21.9 | 50.0 | 34.4 | 40.6 | 34.4 | 41.2 |
| W/ SVC (OURS) | 43.8 | 71.9 | 34.4 | 65.6 | 37.5 | 60.9 | 21.9 | 68.8 | 34.4 | 37.5 | 31.2 | 46.2 (+5.0 ↑) |
| TIES | 40.6 | 56.2 | 34.4 | **71.9** | 37.5 | **68.8** | 21.9 | 59.4 | 34.4 | **43.8** | 31.2 | 45.5 |
| W/ SVC (OURS) | 43.8 | 75.0 | **50.0** | **71.9** | 37.5 | 59.4 | 25.0 | **78.1** | 34.4 | 40.6 | 31.2 | **49.7** (+4.2 ↑) |
| TSV-M | **43.8** | 71.9 | 40.6 | 57.8 | 37.5 | 59.4 | 25.0 | 68.8 | 34.4 | 40.6 | 31.2 | 46.5 |
| W/ SVC (OURS) | **43.8** | 71.9 | 40.6 | 57.8 | 37.5 | 57.8 | 25.0 | 71.9 | 34.4 | 40.6 | 31.2 | 46.6 (+0.1 ↑) |
| ISO-C | 40.6 | 68.8 | 37.5 | 57.8 | 37.5 | 50.0 | **28.1** | 59.4 | 34.4 | 40.6 | 28.1 | 43.9 |
| W/ SVC (OURS) | 43.8 | **78.1** | 43.8 | 68.8 | 37.5 | 62.5 | **28.1** | 68.8 | 34.4 | 40.6 | 31.2 | 48.9 (+5.0 ↑) |
| ISO-CTS | 40.6 | 53.1 | 37.5 | 42.2 | **39.1** | 48.4 | 25.0 | 43.8 | 31.2 | 37.5 | 28.1 | 38.8 |
| W/ SVC (OURS) | 43.8 | 75.0 | 43.8 | 53.1 | 37.5 | 56.2 | **28.1** | 65.6 | **34.4** | 40.6 | 31.2 | 46.3 (+7.5 ↑) |

*Table 15.* Comparison of model merging methods across 11 NLP benchmarks using bigscience/T0_3B (IA3). Bold values indicate the best performance among merging-based techniques. The notation "(Ours)" highlights the integration of our proposed SVC method.

| METHOD | RTE | CB | WINOGR. | WIC | WSC | COPA | H-SWAG | STORY | ANLI-R1 | ANLI-R2 | ANLI-R3 | AVG. |
|---|---|---|---|---|---|---|---|---|---|---|---|---|
| TA | 71.9 | 56.2 | 53.1 | 29.6 | **65.6** | 78.1 | 46.9 | 87.5 | 46.9 | 28.1 | 25.0 | 53.5 |
| w/ SVC | 71.9 | **81.2** | **59.4** | **67.2** | 43.8 | **93.8** | **50.0** | **93.8** | **56.2** | **46.9** | **59.4** | **65.8** (+12.3 ↑) |
| TIES | 71.9 | 59.4 | 53.1 | 31.2 | 62.5 | 79.6 | 46.9 | 87.5 | 46.9 | 31.2 | 25.0 | 54.1 |
| w/ SVC | 71.9 | 59.4 | 53.1 | 29.6 | **65.6** | 78.1 | 46.9 | 78.1 | 46.9 | 31.2 | 25.0 | 54.1 (+0.0 ↑) |
| DARE | 71.9 | 59.4 | 53.1 | 29.6 | 62.5 | 78.1 | 43.8 | 87.5 | 46.9 | 28.1 | 25.0 | 53.3 |
| w/ SVC | **75.0** | 56.2 | 53.1 | 31.2 | 62.5 | 79.6 | 46.9 | 87.5 | 50.0 | 31.2 | 28.1 | 54.7 (+1.4 ↑) |

