# OpenReview forum: "When Shared Knowledge Hurts: Spectral Over-Accumulation in Model Merging"
_ICML.cc/2026/Conference — ICML 2026 regular_

### Official Review · Reviewer_SgPt · 2026-03-10

**Soundness:** 3
**Presentation:** 3
**Significance:** 3
**Originality:** 3
**Overall Recommendation:** 4
**Confidence:** 2

**Summary:**

This paper offers profound insights into a neglected issue in model merging. Traditional model merging methods mainly focus on how to solve the 'interference' or 'conflicts' between different tasks. The author pointed out another problem: the excessive accumulation of domain-invariant knowledge. This can lead to a lack of domain-specific knowledge. To solve this problem, the author proposed SVC, which is a post-processing method that requires no training and no data. Scale the merged singular values according to the degree of overlap to restore the original spectral distribution balance of each task. And the author's method is very effective than previous methods.

**Compliance With Llm Reviewing Policy:**

Affirmed.

**Final Justification:**

My initial concerns centered on the sensitivity of the hyperparameter and the lack of theoretical explanation for the asymmetry between row-space and column-space performance. The authors' rebuttal addressed both points. As my primary concerns have been resolved, and given that this sub-domain is not my area of expertise, I maintain my initial positive assessment.

**Key Questions For Authors:**

See weaknesses.

**Limitations:**

yes

**Strengths And Weaknesses:**

Strengths:

1. Breaking through the traditional framework of 'conflict/interference', it systematically explores for the first time the performance decline caused by 'shared knowledge overload', which is highly enlightening.

2. The paper provides rigorous mathematical derivations, explaining how alignment bias leads to singular value inflation, making the methodology not merely empirical.

3. Efficient and practical: SVC is a typical 'plug-and-play' method, requiring no training or auxiliary data, and has relatively low computational overhead

Weaknesses:

1. Hyperparameters are sensitive. Alpha needs different adjustments for different tasks. The sensitivity of the Alpha parameter reveals that the weight of domain-specific knowledge is not necessarily better the higher it is, and how to manage this balance when balancing domain-invariant singular values and domain-specific singular values can be further explored.

2. Table 3 shows a significant performance drop when applying SVC to the row-space instead of the column-space. The current explanation for this asymmetry is too brief. I recommend adding a deeper theoretical analysis to clarify why task updates align constructively in the output space but destructively in the input space.

---

> ### Author Rebuttal · Authors · 2026-03-31
>
> Thanks for your suggestions. We'll supplement corresponding discussion in our paper.
>
> ## W1: **Hyperparameters Are Sensitive**
>
> As you rightly pointed out, we have placed strong emphasis on the sensitivity of our method to its hyperparameters. To this end, we introduce the hyperparameter **`alpha`** to control whether singular values are amplified, and if so, to what extent, especially for those you mentioned domain-specific components. Where when **`alpha = 1`**, SVC take an conservative setting that will never increase any singular value.
>
> In terms of the performance, **our SVC is relatively stable** with respect to the **`alpha`**：As stated in Lines 377 and 383, in conservative setting that **`alpha = 1`**, SVC still delivers stable and significant performance gains. This observation is also consistent with Reviewer DcoB’s comment: *“When applying SVC to methods such as TSV-M [4], the performance is highly stable regardless of the hyperparameter alpha. This indicates the robustness of the method and the hyperparameter of choice.”*
>
> Notably, the motivation for introducing this hyperparameter comes precisely from our observation that **more domain-specific knowledge is not necessarily better**. In our analysis, domain-specific components across tasks often exhibit substantial conflict. As a result, after projecting the merged model’s singular vectors onto those of the expert models, the resulting projection coefficients can be quite small, which in turn creates a strong tendency to enlarge such domain-specific singular values. However, increasing these singular values may simultaneously drive the merged domain-specific singular vectors farther away, in the $\ell_2$ sense, from the original domain-specific singular vectors. This tension is the fundamental reason why the current version still depends on a hyperparameter.
>
> Since the main focus of this paper is on the study of shared/domain-invariant singular values, we view a more principled treatment of this balance between shared and domain-specific components as an important direction for future work.
>
>
>
> ## W2: **Explanation of Row-Space Choosing**
>
> We thank the reviewer for this insightful question. The key reason for the asymmetry is that column-space response **$(u^r)^\top \Delta W_i$ restrict the output (i.e., hidden feature) on specific direction (same output direction with the merged model), which can be used to calculate our over-accumulation on such output direction**. However, the row-space response $\Delta W_i v^r$ can also probe the response along the input direction $v^r$, the vector $v^r$ itself is a mixed direction merged by multiple tasks. Therefore, it does not provide a meaningful probe for the individual task-specific update $\Delta W_i$.
>
> Specifically, for the column-space response ($a_i^r=(u^r)^\top \Delta W_i$), we have
>
>  $\langle u^r,\Delta h_i\rangle = a_i^r x, \quad \Delta h_i=\Delta W_i x, \quad \forall x$,
>
> where $\Delta h_i$ is the hidden feature induced by task update $\Delta W_i$, and $\langle u^r, \Delta h_i \rangle$ represents the component of $\Delta h_i$ along direction $u^r$. From this perspective, $a_i^r$ can be viewed as the transformation that maps the input $x$ to the output response along direction $u^r$. Therefore, by measuring the cross-task overlap between $a_i^r$ and $a_j^r$, namely $\langle a_i^r, a_j^r \rangle$, we can quantify how similar the two task updates perform along the output direction $u^r$ along **arbitrary same input**. This is exactly the object needed by SVC: in Section 3.2, accumulation is defined through positive cross-task overlap ($\langle a_j^r,a_i^r\rangle>0$), which leads to ($s_i^r>1$) and singular-value inflation.
>
> By contrast, the row-space quantity ($b_i^r=\Delta W_i v^r$) only probes the update along the **single input direction** $v^r$. Indeed, writing ($x=((v^r)^\top x)v^r+x_\perp$), we can obtain
>
> $\Delta W_i x=((v^r)^\top x) b_i^r+\Delta W_i x_\perp$,
>
> which shows that $b_i^r$ captures only biased input slice related mix task input (not the meaningful task-specific input). Table 3 is fully consistent with this mechanism: replacing column-space SVC with row-space SVC largely removes the gains, confirming that **column-space overlap is the appropriate signal for singular-value calibration.**

---

> > ### Author Rebuttal · Reviewer_SgPt · 2026-04-02
> >
> > I appreciate the authors' clarifications, which have resolved the majority of my initial concerns. Accordingly, I will keep my initial positive score.

---

> > > ### Author Response · Authors · 2026-04-02
> > >
> > > We sincerely thank the reviewer for the positive feedback and for recognizing our clarifications.
> > > We are glad that our responses have addressed the concerns.
> > >
> > > We will further incorporate these clarifications into the final version to improve clarity and presentation.

---

### Official Review · Reviewer_DcoB · 2026-03-12

**Soundness:** 2
**Presentation:** 3
**Significance:** 2
**Originality:** 2
**Overall Recommendation:** 5
**Confidence:** 5

**Summary:**

The paper proposes a new method to alleviate the over-accumulation of shared information during model merging. Following this, they show that their method, Singular Value Calibration, outperforms and complements existing model merging methods to achieve a new state-of-the-art performance. By applying their method to simple baselines such as Task Arithmetic, SVC improves the baseline by an impressive 13%. The authors provide empirical evidence supporting the effectiveness of their method and the overaccumulation of information stored in the shared subspaces.

**Compliance With Llm Reviewing Policy:**

Affirmed.

**Final Justification:**

Initially, the authors omitted key Related Works and experiments for a higher number of models merged. However, the authors ran additional experiments and provided a comprehensive comparison against the omitted Related Works, strengthening the positioning of this paper. I believe that the new comparison against existing theoretical results provides strong theoretical grounding and deeper understanding of Model Merging, which is currently lacking in the field. Therefore, the authors have successfully addressed my concerns and I have adjusted my score accordingly.

**Key Questions For Authors:**

The questions are included in the Strengths and Weaknesses above.

**Limitations:**

yes

**Strengths And Weaknesses:**

### **Strengths**

**1. Clear presentation and writing:**

The paper is well-written, easy to grasp and the figures are informative and intuitive. The mathematical notation utilizes the standard notation and the pseudocode is clear and understandable.

**2. Strong Performance:**

SVC improves baselines methods significantly and sets a new state-of-the-art performance for model merging. The approach improves baseline Task Arithmetic by 13%, which is an impressive result.

**3. Hyperparameter Stability:**

When applying SVC to methods such as TSV-M [4], the performance is highly stable regardless of the hyperparameter alpha. This indicates the robustness of the method and the hyperparameter of choice. Do the authors suggest a default hyperparameter to use?

### **Weaknesses**

**1. Lack of Comparison With Key Related Works**:

To the best of my understanding, the paper's main claim regarding the accumulation of shared directions inflating the respective singular values of shared subspaces has been proven in this paper [1], which has also been applied to a wide range of model merging methods. Could the authors please compare their added novelty compared to [1] and highlight this related work or benchmark it? This would be beneficial to understand the differences between the two methods.

**2. Low Number of Tested Tasks**

The authors evaluate up to 14 tasks for CV tasks. Could the authors expand to 20 tasks as well, since model merging performance degradation is most noticeable for a large number of merged models. The respective tasks are presented in TALL Masks [2], [3] and [4].

**3. Computational Overhead**

It would be useful to benchmark the method against Iso-CTS [3], TSV-M [4], and Subspace Boosting [1] to understand the overhead of this method compared to comparable ones. In addition, it would be valuable to see the overall computational overhead when combining SVC with the other mentioned baselines, as shown in Table 1 and 2.



**References:**

[1] Skorobogat, R., Roth, K., and Georgescu, M. I. Subspace-boosted model merging. In arXiv preprint arXiv:2506.16506, 2025.

[2] Wang, K., Dimitriadis, N., Ortiz-Jimenez, G., Fleuret, F., and Frossard, P. Localizing task information for improved model merging and compression. In International Conference on Machine Learning, 2024.

[3] Marczak, D., Magistri, S., Cygert, S., Twardowski, B., Bagdanov, A. D., and van de Weijer, J. No task left behind: Isotropic model merging with common and task-specific subspaces. In International Conference on Machine Learning, 2025.

[4] Gargiulo, A. A., Crisostomi, D., Bucarelli, M. S., Scardapane, S., Silvestri, F., and Rodolà, E. Task singular vectors: Reducing task interference in model merging. In Conference on Computer Vision and Pattern Recognition, 2025.

---

> ### Author Rebuttal · Authors · 2026-03-31
>
> We sincerely appreciate the reviewers for dedicated effort and will revise the paper accordingly.
>
> ## W1: **Comparison With Key Related Works**
>
> Thanks for pointing out **Subspace-Boosted** [1] method. However, **our novelty is not another SVD-based treatment of the same issue, but the identification and correction of a different underexplored failure mode.**
>
> **The starting point is different.**
> Subspace-Boosted focuses on **inter-rank structure**, primarily addressing rank collapse in the lower singular values. In contrast, our SVC focuses on **inter-task interactions**, explaining why even well-aligned tasks can degrade merging performance, where such alignment is mainly reflected in the top singular vectors. Through theoretical analysis, we are the first to reveal that **shared knowledge in dominant spectral subspaces can be over-accumulated**, a critical yet largely overlooked issue, which constitutes our added **novelty**. Fig. 4 shows that this issue is prevalent across a wide range of existing merging methods, including **Subspace-Boosted** [1].
>
> **The calibration principle is also different.**
> Subspace-Boosted method applies an **isotropic** singular value adjustment (rescale them to the same value) to counter collapse. However, our analysis shows that an **anisotropic** adjustment is required, since the over-accumulation of shared knowledge is non-uniformly distributed across subspaces (Fig. 2). This analysis lead to our another **added novelty** compared to [1], our SVC is the first method that emphasizes **subspace-level calibration** rather than uniform spectral treatment. In other words, SVC adjust different the singular value of different subspaces to different target value.
>
> In fact, our experiments suggest that the two methods are **complementary rather than redundant**: applying SVC on top of Subspace-Boosted can resolve its "over-accumulation" issue and yields further gains (see table in our W2 response). Besides, we also added a experiment comparison. With the same fixed α=0.125 as also used in our paper, SVC outperforms Subspace-Boosted with **validation set hyperparameter search** for its  hyperparameter beta:
>
> ||Val.|Task Arithmetic|TIES-Merging|Consensus Merging|Iso-CTS|
> |-|-|-|-|-|-|
> |w/ Subspace-Boost |✔| 76.00%| 77.65%|68.00%|83.35%|
> |w/ SVC|❌|77.07%|84.86%|81.28%|84.14%|
>
> Therefore, **the contribution of SVC is not merely another spectral post-processing method, but a distinct explanation and correction of shared-knowledge over-accumulation in dominant subspaces**. We will revise the paper to make this distinction explicit and to discuss [1] more thoroughly.
>
> ## W2: **More Number of Tested Tasks**
>
> We agree that the 20-task CV setting is a valuable stress test. However, since such large-scale model merging is uncommon in realistic applications, we omitted this in our paper. We provide the additional results below.
>
> We try our best to run a closely matched reproduction of the TALL Masks (Consensus Merge) / TSV-M / Iso-C / Iso-CTS setting (noting that the fine-tuned checkpoints used there are the same with the paper of Iso-CTS, but different from ours paper). Under this reproduced protocol, SVC still remains useful as an add-on across the available baselines. This further supports our claim that the over-accumulation issue addressed by SVC is an important yet largely overlooked problem in model merging. However, due to the increase in the number of tasks, the conflict between the task response $a_{i}^r$ of expert model and merged response $a_{merge}^r$ of merged model become more obvious, leading our post-calibration method is relatively limited compared to the situation with 8 tasks (Iso-CTS 81.4% → 85.6%).
>
> ||Task Arithmetic|Subspace-Boosted|TALL Mask|TSV-M|Iso-C|Iso-CTS|
> |-|-|-|-|-|-|-|
> |w/o SVC|60.98%|68.89%|37.94%|76.63%|75.21%|77.64%|
> |w/ SVC|61.19%|71.47%|54.41%|76.98%|75.90%|77.72%|
>
> **Note:** Our reproduction of TALL-Mask currently underperforms the reported results. Due to rebuttal time constraints, we have not yet identified the cause, but we will investigate this discrepancy and update the results in the revised version.
>
>
> ## W3: **Computational Overhead**
>
> The current paper reports standalone SVC cost (Table 4: 5.1s / 517.2s on ViT-B/32 and LLaMA2-7B). The overall computational overhead =  overhead of `baseline` + the overhead of our `SVC`. The primary results are as follows. It can be seen that our method only brings additional cost of the order of seconds in general, which is completely acceptable, showing our calibration efficiency.
>
> | |ViT-B/32 (8 Tasks)|Llama2-7B|BERT|
> |-|-|-|-|
> |TA|0.2s|52.4s|0.1s|
> |TA w/ SVC|4.7s|532.3s|6.1s|
> |TIES|9.5s|156.6s|3.9s|
> |TIES w/ SVC|13.4s|656.6s|10.0s|
> |DARE|12.5s|184.8s|4.2s|
> |DARE w/ SVC|16.3s|698.7s|9.5s|
> |TSV-M| 28.5s|238.1s|20.9s|
> |TSV-M w/ SVC|32.4s|750.4s|25.9s|
> |Iso-C| 4.2s|263.0s|4.6s|
> |Iso-C w/ SVC|9.6s|769.2s|13.7s|
> |Iso-CTS|123.6s|378.7s|75.1s|
> |Iso-CTS w/ SVC|128.5s|882.5s|82.8s|

---

> > ### Author Rebuttal · Reviewer_DcoB · 2026-04-03
> >
> > I thank the authors for their detailed rebuttal and additional results. The additional results look promising. However, I still have open concerns regarding W1 before I raise my score.
> >
> > W1: "shared knowledge in dominant spectral subspaces can be over-accumulated"
> >
> > From my understanding, the above statement is equivalent to this statement in Subspace-Boosted Model Merging:
> > Proposition 1: "Consequently, the gap between the common subspace and task-specific subspace widens." or "any Task Arithmetic-based method will inevitably emphasize the common components while disregarding the task-specific information as more models are merged".
> >
> > Could the authors explain this point further? As I understand, the common subspace/components in Subspace-Boosted Model Merging corresponds to the dominant spectral subspace in this work. The singular values from the merged task vectors correspond to the tasks themselves, so if I understand correctly, the singular values and tasks are interchangeable here and you argue similarly to the previous work.
> >
> > I look forward to raising the score once this final point is clarified and compared further against [1].
> >
> > [1] Skorobogat, R., Roth, K., and Georgescu, M. I. Subspace-boosted model merging. In arXiv preprint arXiv:2506.16506, 2025.

---

> > > ### Author Response · Authors · 2026-04-05
> > >
> > > Thank you again for this helpful follow-up. We also sincerely thanks for pointing us to **Subspace-Boosted Method**, a relevant work developed independently along the similar research direction with ours. We highly acknowledge its analysis of the common and task-specific subspace, but we would like to clarify that the two claims are **related, but not equivalent**.
> > >
> > > We copy **proposition 3** of Subspace-Boosted below:
> > >
> > > > Let the merged task vector be defined by standard Task Arithmetic as $\Delta_m=\alpha\sum_{i=1}^N\Delta_i$ for any scalar merging coefficient $\alpha\in\mathbb{R}$. Let $\Delta_m=\Delta_{common}+\Delta_{unique}$. As $N → ∞$, the ratio of the spectral magnitude of the common subspace to the task-specific subspace diverges:
> > > > $$
> > > > \frac{\left\|\Delta_{\text{common }}\right\|_2}{\left\|\Delta\_{\text{unique}}\right\|_2}\geq\mathcal{O}(\sqrt{N}).
> > > > $$
> > >
> > > This give us 2 key insights:
> > >
> > > (i) merged model contains both shared and task-specific parts;
> > >
> > > (ii) shared part becomes increasingly dominant as more models are merged.
> > >
> > > This observation is relevant to ours, which provides a **global and qualitative** description of spectral dominance.
> > >
> > > We would like to clarify that our claim is **more specific and more quantitative**. Our work instead asks a finer-grained question: **what exactly is the shared knowledge**, and **why can an increased proportion of shared knowledge hurt the merged model?** In parallel, we give our following proposition:
> > >
> > > > **Proposition (Ours, Shared-Knowledge Over-Accumulation).**
> > > >
> > > > Consider a merged model $\Delta W_\text{merge}$ and one of its output spectral directions $u_{\mathrm{merge}}^{r}$. Define the expert response of task $i$ and the merged response on this output direction as $a_i^{r}=\left(u_{\text{merge}}^{r}\right)^{\top} \Delta W_i$ and $a_{\text{merge}}^{r}=\left(u_{\text {merge}}^{r}\right)^{\top}\Delta W_\text{merge}$. We define the overlap $\langle a_i^{r},a_j^{r}\rangle$ between two expert responses on the **same output direction** $u_{\mathrm{merge}}^{r}$ as the shared knowledge on this direction. Furthermore, define the cross-term between the merged response and expert response as
> > > > $$
> > > > \langle a_\text{merge}^{r},a_i^{r}\rangle=s_i^r||a_i^r||\_2^2
> > > > $$
> > > > where $s_i^r$ is the projection coefficient of $a_{\mathrm{merge}}^{r}$ onto $a_i^{r}$. Then $s_i^r > 1$ indicates that aligned responses are excessively accumulated on $u_{\mathrm{merge}}^{r}$: the merged model responds more strongly on this output direction than what is supported by expert $i$, thereby increasing the output-space interference (i.e., discrepancy between the merged model and the experts in the output space), hurt the merged model.
> > >
> > > In short, compared with Subspace-Boosted Model Merging, which characterizes the phenomenon through a **global decomposition** into common and unique components, our method provides a **quantitative subspace-level explanation**. Specifically, to the best of our knowledge, our paper is the first to explicitly **define and measure shared knowledge through task responses on the same merged output direction**, and to identify subspaces with $s_i^r-1>0$ as directions where shared knowledge is over-accumulated. More importantly, our theory first explains **why increased proportion of shared knowledge is harmful, by showing that over-accumulation in $a_\text{merge}^{r}$ will cause higher $s_i^r$, thereby increasing output-space interference** and thus enlarges the behavioral discrepancy between the merged model and the expert models.
> > >
> > > **Proof. of Our Proposition**
> > >
> > > We first define the **output-space interference** in the merged output subspace $U_{\mathrm{merge}}$ as
> > > $$
> > > I=\sum_{i=1}^K\left\|U_{merge}^\top\Delta W_{merge}−U_{merge}^\top\Delta W_i\right\|\_F^2
> > > $$
> > > This quantity measures the discrepancy between the merged update $\Delta W_{\mathrm{merge}}$ and each expert update $\Delta W_i$ **within the merged model’s output spectral subspace**. It captures their behavioral difference in output space, rather than direct parameter difference. Then:
> > > $$
> > > I=\sum_{i=1}^K\sum_{r=1}^R\left\|\sigma_r\left(v_{\text{merge}}^{r}\right)^{\top}-\left(u_{\text{merge}}^{r}\right)^{\top}\Delta W_i\right\|\_2^2=\sum_{i=1}^K\sum_{r=1}^R\left\|a_{\text {merge}}^{r}-a_i^{r}\right\|\_2^2
> > > $$
> > > Furthermore, by projection, we can decompose $a_{\text{merge}}^{r}$  into the component along $a_{i,\perp}^{r}$ and an orthogonal component $a_{i,\perp}^{r}$:
> > > $$
> > > I=\sum_{i=1}^K\sum_{r=1}^R\left\|(s_i^r-1)a_i^{r}+\gamma a\_{i,\perp}^{r}\right\|\_2^2
> > > $$
> > > where
> > > $$
> > > s_i^r=\frac{\langle a_{\text{merge}}^{r},a_{i}^{r}\rangle}{\left\|a_i^{r}\right\|\_2^2}
> > > $$
> > > When $a_{\text{merge}}^{r}$ contains more shared information aligned with $a_i^{r}$, the corresponding projection coefficient $s_i^r$ becomes larger, and the interference correspondingly increases. Therefore, $s_i^r > 1$ in the interference $I$ as precisely the harmful effect of shared-information over-accumulation on the merged model.

---

### Official Review · Reviewer_z257 · 2026-03-26

**Soundness:** 2
**Presentation:** 2
**Significance:** 2
**Originality:** 3
**Overall Recommendation:** 3
**Confidence:** 4

**Summary:**

This paper focused on addressing the over-counting shared knowledge which may degrade overall performance in model merging. To mitigate this issue, it proposes a training-free and data-free method, singular value calibration (SVC), to quantify subspace overlap across models and rescale inflated singular values, aiming to restore a balanced spectrum. The theoretical analysis motivates and supports the design of SVC. The experimental results show the effectiveness of SVC.

**Compliance With Llm Reviewing Policy:**

Affirmed.

**Final Justification:**

The contribution of SVC remains unclear. If SVC is intended as a post calibration rule, I would expect it to operate solely on a merged model without access to all fine-tuned models. Therefore, I keep my score.

**Key Questions For Authors:**

1. In lines 130 to 132, can authors explain clearly why using $v$ will lose information about how the task interacts with all possible inputs? Is there any theoretical explanation?

2. In Table 1 and Table 2, the experimental results only present SVC combined with other methods, while there is no performance of SVC itself. Can authors explain why SVC cannot be evaluated alone? Showing the combined results can only explain the performance of how the existing methods fit SVC. I think there should be one line which only presents SVC results to help isolate its contribution. Having a reasonable response can help re-evaluate the scoring.

3. Can authors provide the results of Qwen2.5-7B model in Table 4?

**Limitations:**

The evaluation of the proposed method should be well-explained.

**Strengths And Weaknesses:**

Strengths:

1. The optimal calibration analysis in section 3.2 is well-motivated and provides theoretical support for the design of SVC.

2. The proposed method, SVC, is data-free and training-free.

Weaknesses:

1. The paper mainly used Task Arithmetic as a baseline in both the introduction and method sections, but Task Arithmetic is a foundational milestone method in model merging. After that, there are lots of model merging methods, including addressing shared knowledge and subspace overlap [1]. Thus, it is not sufficiently convincing to explain the advantage of the proposed method by using the prior work which did not consider that aspect, but ignoring the similar prior work. I strongly suggest that authors compare SVC with Iso-CTS from the perspective of the method principle, since both methods handle shared and task-specific components. Reasonable responses can help re-evaluate the scoring.

     [1] No Task Left Behind: Isotropic Model Merging with Common and Task-Specific Subspaces, ICML2025

2. There are several statements in the paper that lack reference, especially in the third paragraph of the introduction, which weakens the confidence of the paper. For example, the claim “A task is influenced not only by its task-specific update, but also by shared knowledge introduced by other tasks through the merge.”

3. In Figure 3, I found that the cross-term values shown on the right vertical axis range only from 0.000 to 0.006. Such small values can be seen as almost no overlap, which appears inconsistent with the figure’s title and intended interpretation of Figure 3.

4. Clarification and presentation: (1) Figure 2, Figure 3, and Figure 4 have no description of the experimental setup, making them hard to interpret. (2) In Figure 2 and Figure 4, the color choices make it hard to distinguish the information. (3) In Eq(3), I did not find the definition of $w^{r}$.

---

> ### Author Rebuttal · Authors · 2026-03-31
>
> We sincerely appreciate the reviewers for valuable comments and will revise the paper accordingly.
>
> ## W1: **Relation to Iso-CTS**
>
> Thanks for suggesting a broader comparison with Iso-CTS. Our claim is not that prior work never observed spectral alignment, but that we identify a novel complementary post-merge mechanism: repeated accumulation of shared directions in the merged model inflates dominant singular values.
>
> **First, they differ shared knowledge.**
> Iso-CTS explicitly treats selective top singular directions of the merged model as the shared knowledge, and applies an **isotropic** adjustment to them. From this perspective, shared knowledge is regarded as a uniform whole, lacking fine-grained analysis. In contrast, our analysis reveal that shared knowledge is highly **anisotropic across subspaces** (Fig. 2): top singular directions are strongly aligned (80% similarity) across tasks and carry most shared knowledge, while lower ones are often weakly aligned or even opposed. This anisotropy lead SVC performs **fine-grained, subspace-level calibration** rather than uniform spectral treatment.
>
> **Second, they also differ in task-specific knowledge.**
> Iso-CTS extracts task-specific components **before merging** by calculating the orthogonal parts of expert model weights towards shared directions. In contrast, SVC identifies these components in the merged model **after merging**. By Eq. (19), we find the average projection coefficient of $a_{merge}^r$ onto $a_i^r$ is only `0.03` in bottom subspaces, which often `exceeds 1` in top subspaces. This first reveal that bottom singular directions of the merged model are formed by task-specific components with small similarity that are easily canceled during merging. Thus, unlike Iso-CTS, SVC does not inject task-specific knowledge into the merged model, but **mitigates the excessive cancellation** of task-specific knowledge already present in the merge.
>
> In short, Iso-CTS and SVC **target different failure modes and are therefore distinct yet complementary**. Our experiments support this: Iso-CTS with SVC improves accuracy from 81.4→85.6 in the 8-task setting, and from 43.8 to 51.3 on AlpacaEval (Llama), indicating that shared-knowledge over-accumulation remains even after Iso-CTS balances the spectrum. Given the similarity between Iso-CTS and the Subspace-Boost, we also refer the reviewer to our response to Reviewer DcoB, W1, for a comparison from more general perspectives.
>
> ## W2: **Missing References in the Introduction**
>
> Our intended point is that a merged model contains not only task-specific components, but also shared components introduced during merging, a widely recognized observation in prior work (e.g., [1]). On this basis, we further introduce the over-accumulated issue in such shared components that were overlooked in the previous studies. We will revise the wording and add an appropriate references in the final version .
>
> [1] TIES-Merging: Resolving Interference When Merging Models.
>
> ## W3: **Figure 3 Scale**
>
> Figure 3 shows the raw cross-terms $\langle a_i^r, a_j^r\rangle$, not a normalized similarity. In normalized form, it's **diagonal average increases to about 0.49** (when two index $r$ are the same), which is strong for 768-dimensional vectors, while the **off-diagonal average is only $6.27\times10^{-6}$**. We use raw cross-terms because our analysis relies on their **absolute magnitude**, we will clarify this in the revision to avoid confusion.
>
> ## W4: **Clarification and Presentation**
>
> We will revise **Figures 2–4** to improve readability. For **Eq. (3)**, we apologize for the confusion: $w_k^\top$ denotes the $k$-th row of $\Delta W_i$ (with $w_k \in \mathbb{R}^n$); there is no separate variable $w^r$.
>
> ## W5: **Why ($\Delta W_i v$) Loses Information**
>
> In short, right-multiplication by a single vector $v$ restricts the effect of $\Delta W_i$ to a specific input direction, making the subsequent analysis on $\Delta W_i$ no longer comprehensive. Due to the rebuttal length limit, a more detailed explanation is provided in our response to Reviewer SgPt, W2.
>
> ## W6: **Why SVC Is Not Reported Alone**
>
> As shown in the pseudocode, SVC is a **post calibration rule, not a standalone merge operator**. It operates on an existing merged model to detect and rescale over-accumulated shared directions, aiming to correct this specific failure mode rather than build a merged model from scratch. We will clarify this in the revision.
>
> ## W7: **Qwen2.5-7B in Table 4**
>
> We added additional Qwen2.5-7B overhead experiment to supplement Table 4. When merging two fully fine-tuned Qwen2.5-7B models on a single A100 GPU, SVC adds only **249.3 seconds** of runtime and **2513.1 MB** of GPU memory. Compared with the **hours** required to train a 7B model, this **overhead is negligible**.
>
> |Backbone|Time Cost|Memory Usage|
> |-|-|-|
> |ViT-B/32|5.1s|1027.4MiB|
> |ViT-B/16|8.2s|1082.8MiB|
> |ViT-L/14|15.6s|1488.5MiB|
> |LLaMA2-7B|517.2s|1898.7MiB|
> |Qwen2.5-7B|249.3s|2513.1MiB|

---

> > ### Author Rebuttal · Reviewer_z257 · 2026-04-03
> >
> > Thanks for the authors' response. I appreciate adding new experiments with the Qwen model, which helps strengthen the evaluation. But I still have concerns about the contribution of SVC. The authors describe SVC as "post calibration rule, not a standalone merge operator", but SVC still requires access to each fine-tuned model that is being merged, as indicated in Algorithm 1. In this sense, it's still like a merging operator, since it directly operates on the fine-tuned models. If SVC is a post calibration rule, I would expect it to act on a merged model without access to all fine-tuned models.
> >
> > Therefore, I would keep my original score.

---

> > > ### Author Response · Authors · 2026-04-05
> > >
> > > We sincerely thank the reviewer for this important follow-up. We believe the remaining concern may mainly come from a different interpretation of **post-calibration**. We use this term in the same sense as the recent model merging survey [1], which explicitly summarizes *"Post-calibration-based Merging Methods"* in Table 1 as: ***“Calibrating the merged model to be closer to the individual models reduces the knowledge loss.”*** This statement itself makes clear that accessing the individual models (i.e., fine-tuned models) is a standard and fully accepted design in post-calibration methods, since without the guidance of the individual models, the calibration process itself would lack a sufficient basis for judging whether the current correction direction of the merged model is truly making it closer to the individual models.
> > > Therefore, **Most of existing methods with respect to post calibration rule can access to all fine-tuned models**.
> > >
> > > Our SVC follows exactly this spirit. It does not rely on any extra information beyond the standard model merging setup, because the fine-tuned expert models are already available during merging. Our method uses these expert fine-tuned models only as **references** to measure how the **already merged model** deviates from them in spectral-response space, and then calibrates the merged model accordingly. This is also consistent with **Representation Surgery** (ICML'24) [2], a representative post-calibration / post-merging method, which likewise compares the merged model with the individual models to reduce their representation discrepancy. Like Representation Surgery [2], which is designed on top of another standalone merge operator (e.g., Task Arithmetic w/ Surgery; TIES-Merging w/ Surgery; AdaMerging w/ Surgery), our SVC is likewise built on top of existing merge methods to perform calibration (e.g., Task Arithmetic w/ SVC; TIES-Merging w/ SVC; TSV-M w/ SVC; Iso-CTS w/ SVC). Moreover, as demonstrated by extensive experiments across different architectures, **the effectiveness of our method does not rely on one or two specific merge operators. Instead, its gains come from reducing a generally existing gap between the merged model and the expert models**.
> > >
> > >
> > > More importantly, we argue that our method, as well as other similar post-calibration-perspective methods, also makes an **important contribution** to the model merging field: our SVC not only serves as a **plug-and-play** module that can improve the performance of existing, and potentially future, merging methods, but also reveals **where** the merged model differs from the expert models, and such findings can in turn guide the future design of better standalone merge operators. Specifically, our SVC reveals a previously underexplored discrepancy between the merged model and the fine-tuned expert models: after merging, aligned shared responses can become **over-accumulated** in dominant spectral subspaces. In essence, our SVC can be viewed as implicitly associated with an optimization objective:
> > >
> > > $$
> > > \min\sum_{i=1}^K \left\| U_{merge}^\top \Delta W_{merge} − U_{merge}^\top \Delta W_i \right\|_F^2,
> > > $$
> > >
> > > $$
> > > = \min \sum_{i=1}^K \sum_{r=1}^R\left\|\left(u_{\text {merge}}^{r}\right)^{\top} \Delta W_{\text {merge}} - \left(u_{\text {merge}}^{r}\right)^{\top} \Delta W_i \right\|_2^2
> > > $$
> > >
> > > $$
> > > = \min \sum_{i=1}^K \sum_{r=1}^R\left\|\sigma_r\left(v_{\text{merge}}^{r}\right)^{\top}    - \left(u_{\text {merge}}^{r}\right)^{\top}\Delta W_i\right\|_2^2
> > > $$
> > >
> > > In our post-calibration setting, we keep the direction of singular vectors $U_{merge}$, and only optimize the easily handle part $\sigma_r$. This may also provide useful guidance for future standalone merge operators: if the direction of $U_{merge}$  were also allowed to change, could one obtain an even better merged model from standalone merge view?
> > >
> > >
> > >
> > > **References**
> > >
> > >  [1] Yang et al., *Model Merging in LLMs, MLLMs, and Beyond: Methods, Theories, Applications and Opportunities*, arXiv, 2024.
> > >
> > >  [2] Yang et al., *Representation Surgery for Multi-Task Model Merging*, ICML 2024.

---

### Decision · Program_Chairs · 2026-04-30

**Decision:**

Accept (regular)

**Comment:**

This paper proposes SVC, a training-free and data-free post-processing technique for model merging. Reviewers broadly appreciated the paper’s conceptual contribution. In particular, they welcomed the paper’s new perspective on model merging, and most concerns were resolved during the rebuttal. One unresolved concern from a reviewer relates to the use of the term “post-calibration.” The paper claims this characterization, yet SVC requires access to all individually fine-tuned models during its operation, which makes it function more like a merging operator than a true post-calibration rule. The authors responded that such access is standard practice in the post-calibration literature and is consistent with how recent surveys and representative methods, such as Representation Surgery, define the category. While this explanation is reasonable, the terminology as currently used in the paper may still cause confusion for readers unfamiliar with this convention and should be clarified in a revised version. The authors are strongly encouraged to incorporate all discussions, results, and clarifications from the rebuttal into the subsequent version to improve the paper’s clarity and overall quality.